



# 1 The impact of lateral boundary forcing in the CORDEX-Africa
ensemble over southern Africa
Maria Chara Karypidou[1], Stefan Pieter Sobolowski[2], Eleni Katragkou[1], Lorenzo Sangelantoni[3,4], Grigory Nikulin[5]
[1]Department of Meteorology and Climatology, School of Geology, Faculty of Sciences, Aristotle University of
Thessaloniki, Thessaloniki, Greece
[2]NORCE Norwegian Research Centre, Bjerknes Centre for Climate Research, Bergen, Norway
[3]CETEMPS—Department of Physical and Chemical Sciences, University of L'Aquila, L'Aquila, Italy
[4]Department of Physical and Chemical Sciences, University of L'Aquila, L'Aquila, Italy
[5]Rossby Centre, Swedish Meteorological and Hydrological Institute, Norrköping, Sweden
Corresponding author: Maria Chara Karypidou, karypidou@geo.auth.gr

## 13 Abstract

The region of southern Africa (SAF) is among the most exposed climate change hotspots and is projected to experience
severe impacts on multiple economical and societal sectors. For this reason, producing reliable projections of the
expected impacts of climate change is key for local communities. In this work we use a set of 19 regional climate
models (RCMs) performed in the context of the Coordinated Regional Climate Downscaling Experiment (CORDEX)
– Africa and a set of 10 global climate models (GCMs) participating in the Coupled Model Intercomparison Project
Phase 5 (CMIP5), that were used as the driving GCMs in the RCM simulations. We are concerned about the degree
to which RCM simulations are influenced by their driving GCMs, with regards to monthly precipitation climatologies,
precipitation biases and precipitation change signal, according to the Representative Concentration Pathway (RCP)
8.5 for the end of the 21st century. We investigate the degree to which RCMs and GCMs are able to reproduce specific
climatic features over SAF and over three sub-regions, namely the greater Angola region, the greater Mozambique
region and the greater South Africa region. We identify that during the beginning of the rainy season, when regional
processes are largely dependent on the coupling between the surface and the atmosphere, the impact of the driving
GCMs on the RCMs is smaller, compared to the core of the rainy season, when precipitation is mainly controlled by
the large-scale circulation. In addition, we show that RCMs are able to counteract the bias received by their driving
GCMs, hence, we claim that the cascade of uncertainty over SAF is not additive, but indeed the RCMs do provide
improved precipitation climatologies. The fact that certain bias patterns over the historical period (1985-2005)
identified in GCMs are resolved in RCMs, provides evidence that RCMs are reliable tools for climate change impact
studies over SAF.



## 1 Introduction

The region of southern Africa (SAF) is among the most exposed climate change hotspots (Diffenbaugh and Giorgi, 2012), and is projected to experience severe impacts on multiple economical and societal sectors (Conway et al., 2015; Masipa, 2017; Shew et al., 2020). Poverty, food insecurity and high levels of malnutrition (Misselhorn and Hendriks, 2017) render SAF a region particularly vulnerable to the impacts of climate change (Casale et al., 2010; Luan et al., 2013; Wolski et al., 2020). In addition, the population's reliance on rain-fed agriculture makes strategic planning necessary as it aims to mitigate the impact of climate change on local communities.

Global climate models (GCM) participating in the Coupled Model Intercomparison Project Phase 5 (CMIP5) (Taylor et al., 2012) project a significant decline in annual precipitation over SAF (IPCC and Stocker, 2013), with the most pronounced changes projected under representative concentration pathway 8.5 (RCP8.5) (Sillmann et al., 2013). This reduction is also identified in the regional climate model (RCM) simulations performed in the context of the Coordinated Regional Climate Downscaling Experiment (CORDEX) – Africa domain (Nikulin et al., 2012; Giorgi and Gutowski, 2015). More specifically according to CORDEX-Africa simulations, annual precipitation is expected to decline by up to 50% by the end of the 21$^{st}$ century (Pinto et al., 2018), while duration of dry spells is projected to increase (Dosio et al., 2019). Despite this, extreme rain events are expected to increase in frequency and intensity (Pinto et al., 2016; Abiodun et al., 2019). Nevertheless, for a global warming level of 2 °C, certain parts of SAF (northern Angola, Zambia, northern Mozambique and eastern South Africa) are projected to experience precipitation increase during specific times of the year (Maúre et al., 2018).

The question of whether or not RCMs produce demonstrable added value relative to their driving GCMs, has often fueled debate between the RCM and GCM modelling communities (Lloyd et al., 2020). The outcome of the debate is not binary. The literature provides ample evidence that there is indeed evidence of added value in RCMs, but it is dependent on the region examined, on the season and the climate mechanisms that are at play (Luca et al., 2016, Feser et al., 2011). RCM ensembles such as those in CORDEX-Africa endeavor to provide added value, by dynamically downscaling historical and scenario simulations originating from coarse resolution GCMs (Dosio et al., 2019). The added value in RCM simulations arises as a result of their higher horizontal resolution (<50 km), which makes it possible for atmospheric waves and synoptic scale disturbances to be represented in a more realistic manner. An additional aspect that further contributes towards this end, is the more accurate representation of land surface characteristics (topography, land use etc.) in RCMs (Di Luca et al., 2013). Moreover, the physics of a RCM can be targeted for processes specific to the region it is being run for, giving it a local advantage over GCMs that may have had their physics developed for global applications. Nevertheless, RCMs also are accompanied by a set of model deficiencies of their own that affect the final output of the downscaled data (Boberg and Christensen, 2012). In Sørland et al. (2018) it is reported that although RCM biases are affected by the driving GCMs, they are nonetheless not additive, a result that counters the common "cascade of uncertainty" criticism. Still, uncertainty arising from both the driving GCM and the downscaling RCM affect the final product, and it is important to diagnose the sources and causes of these errors (Déqué et al., 2012).



Attributing this uncertainty into its respective components is key for a better assessment of the reliability of RCM
simulations (Christensen and Kjellström, 2020). GCMs provide the lateral boundary conditions to the RCMs and each
RCM receives, absorbs, and modulates the received atmospheric forcing in different ways, depending on the numerical
formulations and parameterization schemes employed. Discerning between the signal received by the GCM and the
signal produced by the RCM is critical for assessing the robustness with which different modelling systems are able
to accurately reproduce observed climatologies and generate reliable estimates of the expected climate change. In
addition, the manner in which an RCM responds to the atmospheric forcing provided by a GCM can be region specific
(Rana et al., 2020; Wu and Gao, 2020) (e.g., regions located in close proximity to the boundaries of the RCM domain
can be more severely affected by the driving GCMs, than regions at the center of the RCM domain or there can be
region specific response around complex topography versus lowlands). Also, the degree to which an RCM is
influenced by the driving GCM can be process specific. For instance, when there is a strong large-scale circulation
signal that is introduced to an RCM domain (e.g. advective mid-latitude storms), it is quite likely that the RCM will
be able to reproduce the information that is received at its lateral boundaries. If, however, the large-scale forcing is
weak, then the atmospheric conditions simulated within the RCM domain are more dependent on the dynamic and
thermodynamic processes employed by the RCM (e.g. convective thunderstorms).
In this work we aim to assess whether it is the RCMs or their driving GCMs that dominate monthly precipitation
climatology, monthly precipitation bias and climate change signal over SAF. We take into account the region-specific
characteristics of this question by analyzing SAF and three subregions, namely southeastern Angola, Mozambique
and South Africa. We also consider the different atmospheric processes that are in play over each region by analyzing
monthly climatologies. Precipitation over SAF results from various atmospheric processes that are highly variable
during the rainy season (Oct-Mar), so by performing the analysis on a monthly basis, we are able to indirectly study
how certain processes are reproduced by GCM and RCM simulations. In order to differentiate between the signal
emanating from the RCMs and their driving GCMs, we use the analysis of variance (ANOVA) in both the GCM and
the RCM ensembles (Déqué et al., 2007, 2012). Since the information provided by RCMs will eventually be used by
both climate and non-climate scientists, especially in light of climate change impact studies, we aim to provide some
information with regards to how much each RCM output is affected by its driving GCM and what climate change
signals are identified consistently in both RCMs and GCMs.

## 2 Material and methods

### 2.1 Data

The data analyzed in the current work are displayed in **Table 1** and consist of RCM simulations performed in the
context of CORDEX-Africa, a set of simulations performed in the context of CMIP5, and the CHIRPS satellite rainfall
product (Funk et al., 2015). More specifically, the CORDEX-Africa simulations selected are those that were driven
by more than two GCMs and for which there are runs for both the historical and the future period under RCP8.5. The
CMIP5 GCMs selected are the ones that were used to drive the CORDEX-Africa simulations. All RCM and GCM
simulations were retrieved from the Earth System Grid Federation (https://esgf-data.dkrz.de/projects/esgf-dkrz/). The



CHIRPS rainfall product is used for calculating precipitation biases in both the CORDEX-Africa and CMIP5
ensembles and was retrieved from: https://www.chc.ucsb.edu/data/chirps. CHIRPS is available at 5 km spatial
resolution and for the calculation of biases it was remapped to the coarser resolution grid using conservative
remapping.

Our analysis is split into two sections: the qualitative and the quantitative part. In the qualitative part, we aim to
identify if RCMs exhibit systematic behavior relative to their driving GCMs. For the quantitative part, we aim to
quantify the degree to which monthly precipitation climatologies, biases and climate change signals are affected by
the downscaled RCMs or by the GCMs driving the RCM simulations. For this purpose, we employ an ensemble of 19
RCM simulations driven by 10 GCMs and the driving GCMs that were used to provide the lateral boundary conditions
to the RCMs. From the historical simulations we use the period 1985-2005 and from the projection simulations we
use the period 2065-2095 under RCP8.5. All CORDEX-Africa simulations are available at ~50 km horizontal
resolution, while the horizontal resolution for the driving GCMs is provided in **Table 2**.

**Table 1** Input RCM and GCM simulations used. The CORDEX-Africa simulations are given in the columns. The
CMIP5 GCMs used as driving fields are given in the rows.

|  | CCLM4-8-17.v1 | RCA4.v1 | REMO2009.v1 |
|---|---|---|---|
| **CanESM2** |  | √ |  |
| **CNRM-CM5** | √ | √ |  |
| **EC-EARTH** | √ | √ | √ |
| **HadGEM2-ES** | √ | √ | √ |
| **MIROC5** |  | √ | √ |
| **MPI-ESM-LR** | √ | √ | √ |
| **IPSL-CM5A-LR** |  |  | √ |
| **IPSL-CM5A-MR** |  | √ |  |
| **CSIRO-Mk3-6-0** |  | √ |  |
| **GFDL-ESM2M** |  | √ |  |
| **NorESM1-M** |  | √ |  |

**Table 2** Horizontal resolution of the CMIP5 GCMs used as driving fields in the CORDEX-Africa simulations.

| GCMs | Latitude Res. | Longitude Res. | References |
|---|---|---|---|
| **CanESM2** | 2.7906 $^{\circ}$ | 2.8125 $^{\circ}$ | (CCCma, 2017) |
| **CNRM-CM5** | 1.40008 $^{\circ}$ | 1.40625 $^{\circ}$ | (Voldoire et al., 2013) |
| **CSIRO-Mk3-6-0** | 1.8653 $^{\circ}$ | 1.875 $^{\circ}$ | (Jeffrey et al., 2013) |
| **EC-EARTH** | 1.1215 $^{\circ}$ | 1.125 $^{\circ}$ | (Hazeleger et al., 2010) |
| **GFDL-ESM-2M** | 2.0225 $^{\circ}$ | 2.5 $^{\circ}$ | (Dunne et al., 2012) |
| **HadGEM2-ES** | 1.25 $^{\circ}$ | 1.875 $^{\circ}$ | (Collins et al., 2011) |
| **IPSL-CM5A-MR** | 1.2676 $^{\circ}$ | 2.5 $^{\circ}$ | (Dufresne et al., 2013) |
| **IPSL-CM5A-LR** | 1.894737 $^{\circ}$ | 3.75 $^{\circ}$ |  |
| **MIROC5** | 1.4008 $^{\circ}$ | 1.40625 $^{\circ}$ | (Watanabe et al., 2010) |
| **MPI-ESM-LR** | 1.8653 $^{\circ}$ | 1.875 $^{\circ}$ | (Giorgetta et al., 2013) |
| **NorESM1-M** | 1.894737 $^{\circ}$ | 2.5 $^{\circ}$ | (Bentsen et al., 2013) |





**2.2 Methods**
The study region and subregions considered are depicted in **Fig. 1**. The subregions are selected based on particular
phenomena and processes that are of importance for the seasonal cycle of precipitation. More specifically, Region A
encompasses the entire SAF region and is defined as the area extending from 10 ºE to 42 ºE and from 10 ºS to 35 ºS.
Region B was selected to capture the main region of interest with regards to the Angola Low (AL) pressure system
(Howard and Washington, 2018) and covers the area extending from 14 ºE to 25 ºE and from 11 ºS to 19 ºS. Region
C covers the eastern coastline, Mozambique and surrounding countries and extends from 31 ºE to 41 ºE and from 10
ºS to 28 ºS. Lastly, we define Region D, which covers  much of South Africa and extends from 15 ºE to 33 ºE and
from 26 ºS to 35 ºS.
One of the primary synoptic scale features controlling precipitation over SAF is the Angola Low (AL) pressure system
(Reason and Jagadheesha, 2005; Lyon and Mason, 2007; Crétat et al., 2019; Munday and Washington, 2017; Howard
and Washington, 2018), which has a distinct seasonal cycle throughout the rainy season (Oct-Mar). This motivates its
selection as a subregion for our study. The AL exhibits heat low characteristics during Oct-Nov and tropical low
characteristics during Dec-Feb (Howard and Washington, 2018). This suggests that during Oct-Nov, since
precipitation is thermally induced and thus tightly dependent on land-atmosphere interactions, it will be the RCMs
that are dominant in controlling precipitation processes. As the rainy season progresses, the AL changes to a tropical
low pressure system and its formation is controlled  by the large-scale circulation that is  characterized by easterly
winds from the Indian Ocean that enter SAF via the Mozambique channel. Since precipitation during Dec-Feb is
caused by transient low-pressure systems, we hypothesize that the impact of the driving GCM fields during Dec-Feb
is enhanced.
In addition, the wider area of Mozambique is a region where the majority of tropical cyclones/depressions make
landfall over continental SAF. The occurrence of transient low-pressure systems is enhanced during the core of the
rainy season (Dec-Feb) and thus we are interested in identifying whether the impact of the driving GCMs is dominant
during Dec-Feb. Also, since according to (Muthige et al., 2018), the number of landfalling tropical cyclones under
RCP8.5 is expected to decline in the future, we are interested in examining whether the impact of the driving GCMs
to the RCM simulations will be altered under future conditions. Hence, Region C is used as a region indicative of the
landfalling tropical cyclones/depressions. Lastly, we examine the area encompassing South Africa (Region D) due to
its strong land-ocean gradients, complex topography and strong seasonal variations in rainfall zones.

**2.2.1 Monthly precipitation climatology and bias**
In order to assess whether or not the RCMs improve the monthly precipitation climatologies relative to their driving
GCMs, we employ a method initially described in Kerkhoff et al. (2015) and later employed by Sørland et al. (2018),
which displays in a scatterplot form the RCM increment as a function of the GCM bias. More specifically, the RCM
increment is described as the difference of each RCM simulation from its driving GCM (RCM-GCM). The RCM
increment is plotted against the GCM bias (GCM-OBS). This plot displays whether or not the RCM increment
counteracts the GCM bias. If the RCM increment reduces the GCM bias, then points are expected to lie along the y=-
x line (negative correlation). On the contrary, if the RCM increment increases the GCM bias, then points are expected





to lie along the y=x line (positive correlation). If the RCM increment and the GCM bias are independent, then points
are expected to be scattered randomly.

**2.2.2 Climate change signal**
The climate change signal (CCS) is identified as the monthly mean difference between the future period (2065-2095)
minus the historical period (1985-2005). As an exploratory method of inspecting the differences between each RCM
simulation from its respective driving (GCM) for monthly precipitation during both the historical and the future period,
we subtract the downscaled precipitation field ($RCM_{DRI}$) from its driving (*DRI*), as in **Eq. 1**:

$$DIFF = RCM_{DRI} - DRI \qquad \textbf{Eq. 1}$$

If *DIFF>0*, then we assume that the RCM enhances precipitation, relative to its driving GCM, while if *DIFF<0* then
we assume that the RCM reduces precipitation, relative to its driving GCM. This method is employed in the qualitative
part of the analysis.

**2.2.3 Analysis of variance**
Additionally, we employ an ANOVA decomposition (Déqué et al., 2007, 2012), in order to understand whether it is
the RCMs or their respective driving GCMs that are responsible for controlling precipitation over the historical (1985-
2005) period and the future period (2065-2095). For this purpose, we use two quantities, namely the "inter-RCM"
variance and the "inter-GCM" variance, as in (Déqué et al., 2012). More specifically, the "inter-RCM variance" is the
variance between all the RCM simulations that are driven by the same GCM. Subsequently, all variances obtained for
all driving GCMs are averaged.

$$RCM_{var} = \frac{1}{N_{RCM}} \Sigma_{RCM_j} \left(P_{ij} - \underline{P_j}\right)^2 \qquad \textbf{Eq. 2}$$

The quantity $P_{ij}$ is the monthly precipitation obtained from all RCMs (*j*) that were driven by the same GCM (*i*). The
quantity $P_j$ is the mean monthly precipitation obtained by all RCMs (*j*) that share a common driving GCM (*i*). As a
final step, the average of all variances is calculated.

$$Inter\_RCMvar = \frac{\sum GCM_I}{N} \qquad \textbf{Eq. 3}$$

Similarly, the "inter-GCM" variance describes the variance between all the GCMs that were used to drive a single
RCM and then averaged over all the variances obtained for all driven RCMs.

$$GCM_{var} = \frac{1}{N_{GCM}} \Sigma_{GCM_i} \left(P_{ij} - \underline{P_i}\right)^2 \qquad \textbf{Eq. 4}$$

Likewise, the average of all variances is calculated.

$$Inter\_GCMvar = \frac{\sum RCM_i}{N} \qquad \textbf{Eq. 5}$$

Both "inter-RCM" and "inter-GCM" variances are normalized by the total variance obtained for all months, as in
(Vautard et al., 2020), so that all values, both for historical and projection runs and RCM and GCM simulations are
comparable. A schematic of the process described above is provided in **Fig. S1**.





**3 Results**

The October and January precipitation climatologies for the period 1985-2005 are displayed in **Fig. 2** and **Fig. 3**, respectively. We use October and January climatologies, because these 2 months may be considered representative of the distinctive processes controlling precipitation over SAF (see section 2.2). We avoid using seasonal means, since the temporal averaging of precipitation often obscures attributes that are better identified on a monthly level. The remaining months of the rainy season are shown in the supplementary material. More specifically, we use October as it is the month that heralds the onset of the rainy season and is often associated with weak precipitation and convective processes that are mainly due to excess surface heating. Also, it is during October that the most intense formations of the heat low expression of the AL are observed. Likewise, we use January as it represents the core of the rainy season, with very strong large-scale precipitation, mainly from the southeastern (SE) part of SAF, through transient synoptic scale low pressure systems.

As it is displayed in **Fig. 2**, precipitation during October occurs in the northwestern (NW) part and the SE part of SAF. Precipitation in the NW part is associated with the southward migration of the rainband (Nicholson, 2018), while precipitation over the SE part is associated with an early formation of the tropical temperate troughs (TTTs). As it is evident from **Fig. 2**, CCLM4-8-17.v1 reduces precipitation amounts (approximately 4-5 mm/d) in both the NW and SE parts of SAF, relative to the lateral boundary forcing it receives. On the contrary, RCA4.v1 systematically enhances precipitation amounts, regardless of the driving GCM. Also, precipitation according to RCA4.v1 displays a very localized spatial pattern with very strong spatial heterogeneity. This may be attributed to the fact that the topography is not smoothed enough and leads to high precipitation values over grid boxes with high elevation (Van Vooren et al., 2019). This is particularly evident in the mountainous region over coastal Angola. REMO2009.v1 also enhances precipitation amounts regardless of the driving GCM, however in a much more spatially homogeneous way than RCA4.v1.

As it is shown in **Fig. 3**, high precipitation amounts during January are observed over the northern and eastern regions of SAF. During January, differences among the driving GCMs become more pronounced, however, all models agree on the dry conditions observed over the southwestern (SW) part of SAF. With regards to the downscaled products, CCLM4-8-17.v1 produces high precipitation amounts over the central part of northern SAF but displays varying amounts of precipitation over the coastal parts, depending on the driving GCM. RCA4.v1 downscales precipitation in a very localized pattern and enhances precipitation over areas with steep terrain. Also, precipitation over the lake Malawi region is particularly enhanced, regardless of the driving GCM. REMO2009.v1 displays similar precipitation amounts to its driving GCMs, however it enhances precipitation over the coastal part of Angola and Mozambique and yields excess precipitation over lake Malawi, when it is driven by HadGEM2-ES and IPSL. The monthly climatologies for the rest of the rainy season months are shown in the supplementary material (**Fig. S2** – **S5**).





In **Fig. 4** the monthly precipitation bias for October over SAF is shown. Biases are calculated using the CHIRPS
satellite rainfall product as a reference. With the exception of IPSL-CM5A (LR/MR) and CanESM2, all other GCMs
display a consistent wet bias that ranges from 0.1 – 30 mm/d (in isolated areas), with most values over SAF falling
0.1-3 mm/d. Overall, the same pattern generally holds for RCA4.v1 and REMO2009.v1, while CCLM4-7-18.v1
displays a systematic dry bias that reaches 2 mm/d, when forced with EC-EARTH, MPI-ESM-LR and HadGEM2-ES.
More specifically, concerning RCA4.v1, the region where the highest wet bias is observed is over Region B (the
Angola Low region) and over the NW parts of coastal Angola. The dry bias regions in RCA4.v1 are identified over
the northeastern (NE) and southern parts of SAF and they rarely exceed -1.5 mm/d.
The monthly precipitation biases for January over SAF are shown in **Fig. 5**. There is a prevailing wet bias identified
in almost all GCMs that typically reaches 3 - 3.5 mm/d, however, in MIROC5, NorESM and GFDL-ESM2M the
biases exceed 5 mm/d over a major part of SAF. Another feature that systematically appears in GCMs is a dry bias
over the NE part of SAF. This bias pattern is also identified in almost all RCMs with a systematic wet bias over central
and western SAF and a region of dry bias in the NE part. More specifically, in RCA4.v1 and REMO2009.v1, there is
a dry bias over the NE and the southern coast of SAF, while in CCLM4-7-18.v1 the dry bias over the eastern region
extends inland to cover almost the whole of Mozambique. Another interesting feature is identified around the Angolan
coast, where wet biases exceed 5 mm/d, while over an adjacent region there is a strip of dry biases that reaches 2
mm/d. Considering the abrupt increase in elevation and the steep escarpment over the coastal Angola-Namibia region,
this is possibly caused by local circulation driving excess moisture transport from the Atlantic Ocean and overly
aggressive orographically triggered precipitation on the windward side of the topography (wet bias strip), that leads
to dry conditions in the lee side (dry bias strip) (Howard and Washington, 2018). It is noted that the wet bias over the
coastal region is identified in most of the RCA4.v1 simulations and in all REMO2009.v1 simulations, however, the
dry bias in the lee side is seen in CCLM4-7.18.v1 only. The monthly precipitation biases for the rest of the rainy
season months is shown in the supplementary material (**Fig. S6** – **S9**).

A more detailed look into specific subregions over SAF where certain climatological features and processes are at
play, can help gain a more in-depth insight of how the precipitation biases are distributed during each month of the
rainy season and whether or not the RCMs display any improvement relative to their driving GCMs. For this reason,
we plot the RCM increments (RCM-GCM) as a function of the GCM biases (GCM-OBS). The results for October
over SAF and the 3 subregions are displayed in **Fig. 6**. In general, all points are identified close to the y=-x line, hence
there is a tendency that RCMs systematically counteract GCM biases. There are nonetheless substantial differences
between the four regions. For instance, over Region A (SAF region) the IPSL-MR GCM has a wet bias equal to almost
1 mm/day, which is counteracted by RCA by an increment of -0.4 mm/month. Other RCA simulations when driven
by HadGEM2-ES, CNRM-CM5 or EC-EARTH, display an RCM increment similar to that of the GCM bias, hence
RCMs mitigate the GCM bias. Over Region B (Angola Low region) most of the RCMs display an RCM increment
that is nearly equal to the GCM bias. Similar conclusions are drawn for Regions C and D also. The RCM increments
as a function of the GCM biases for January are shown in **Fig. 7**. For all regions except Region D (South Africa) points



are lying closely to the y=-x line, hence overall, RCM increments counteract the GCM biases. The scatterplots for the
rest of the months of the rainy season are shown in the supplementary material (**Fig. S10 – S13**). In general, although
precipitation in RCMs is strongly dependent on the driving GCMs, the RCM increments are anticorrelated to the GCM
biases. The anticorrelations are particularly strong for the Dec-Mar period of the rainy season over Region A, B and
C, but not over D (**Fig. S14**).
In **Fig. 8** the mean analysis of variance of all RCMs driven by the same GCM and of all GCMs driving the same RCM
is shown. Values are spatially averaged for southern Africa and the 3 subregions examined (land pixels only) and refer
to the period 1985-2005. In Region A, monthly precipitation during October and November is dominated by the
RCMs, while during Jan-Mar, it is the GCMs that play a dominant role in formulating precipitation over SAF. This is
indicative of the impact that RCMs exert on the formulation of precipitation during Oct-Nov-Dec and the fact that the
contribution from the GCMs becomes dominant during Jan-Feb-Mar. The fact that the contribution of RCMs during
Oct-Nov-Dec dominates can be attributed to the fact that precipitation during these months is the result of regional
processes that are largely dependent on the coupling between the surface and the atmosphere. The land-atmosphere
coupling is a characteristic resolved by the RCMs, through mechanisms described in land surface models, planetary
boundary layer schemes, convection schemes etc., making the contribution of the large scale drivers from the GCM
less important. However, during Jan-Feb-Mar we observe that the contribution from the RCMs is reduced, and it is
the GCMs that control the monthly precipitation variability. This can be attributed to the fact that during Jan-Feb-Mar
it is the large-scale circulation that modulates precipitation over SAF and the GCMs control the transient synoptic
scale systems that enter SAF. In Region B, the pattern is similar, however, October and November precipitation are
closer to the diagonal, indicating an almost equal contribution by both RCMs and GCMs. Also, Dec-Feb move closer
to the diagonal, nevertheless, precipitation during March is mainly formulated by GCMs. In Region C, October
remains equally influenced by both RCMs and GCMs, however November and December are dominated by the
influence of the RCMs. In Region D, precipitation for all months except October is influenced by GCMs.
In **Fig. 9** the climate change signal for October precipitation over SAF is depicted. All GCMs agree that October
precipitation  will decline by approximately 2 mm/d over the regions that experience precipitation during this period,
namely the NW and SE parts of SAF. In addition, some GCMs display a minor precipitation increase (0 - 0.5 mm/d)
in the SW part of SAF, while some others display a slightly larger (1.5 mm/d) precipitation increase over the eastern
parts of South Africa. Moreover, it is seen that the precipitation change signal is replicated by almost all the
downscaling RCMs, nevertheless, there are some considerable differences between the RCMs and their driving GCM.
More specifically, RCA4.v1 in almost all simulations, displays a larger reduction of the precipitation change signal
relative to its driving GCM, both in magnitude and in spatial extent. Precipitation changes in CCLM4-8-17.v1 seem
to follow closely the driving GCMs, with a severe exception when CNRM-CM5 is used (the NW part of SAF
experiences precipitation decline almost 4 mm/d larger than in the driving GCM). The case for when CCLM4-8-17.v1
is driven by CNRM-CM5 may be partly caused by the fact that the historical simulation had erroneously used lateral
boundary conditions from a different simulation member of CNRM-CM5 (Vautard et al., 2020). In REMO2009.v1, a
precipitation decline region is identified in the NW part of SAF and a minor precipitation increase over eastern South





299 Africa is identified. This pattern for REMO2009.v1 appears to be consistent, regardless of the driving GCM, which

300 could be partly explained by the fact that precipitation during October is thermally driven, and thus the impact of the

301 driving GCMs is not dominant. The precipitation increase in the SE part of SAF is seen over a localized region and

302 could be associated with an increase in the precipitation caused by the Tropical Temperate Troughs (TTTs) (Ratna et

303 al., 2013; Macron et al., 2014; Shongwe et al., 2015).

304 In **Fig. 10** the climate change signal for precipitation during January is displayed. The precipitation change displays a

305 very strong regional heterogeneity. It is also observed that although there is a strong precipitation change signal in all

306 driving GCMs, not all RCMs downscale the signal uniformly. It is also notable that, even among the GCMs, there are

307 substantial differences in the spatial extent and sign of the change. Nevertheless, there are some features that appear

308 in most of the simulations. For instance, almost all GCMs project drying conditions over the SW part of SAF,

309 especially the coastal zone. The precipitation decline is equal to -1 mm/d. This could be explained by a consistent

310 increase in frequency of the Benguela Coastal Low-Level Jet events (Lima et al., 2019; Reboita et al., 2019), causing

311 oceanic upwelling and a subsequent reduction in precipitation. In addition, there is a subset of GCMs that identify a

312 severe precipitation decline over the Angola region that reaches -5 mm/d. Furthermore, in many GCMs a region of

313 precipitation increase is identified, extending from central SAF towards SE SAF. This is particularly identifiable in

314 HadGEM2-ES, and the RCM simulations forced by it. The monthly precipitation changes for the rest of the rainy

315 season months is shown in the supplementary material (**Fig. S15** – **S18**).

316 In **Fig. 11** the spatial average of the $RCM_{DRI}$ – DRI difference (DIFF) is shown for the whole of SAF (land pixels

317 only). If DIFF>0, it indicates that the RCMs enhance precipitation relative to their driving GCM, while if DIFF<0

318 then RCMs reduce precipitation relative to their driving GCM. As it is shown, DIFF values for October are symmetric

319 around zero and do not exceed the range (-1) – 1 mm/d, either for the historical or the future period. Almost symmetric

320 are the DIFF values for November also, however, their spread increases, reaching values that range (-2) – 2 mm/d. In

321 both months, CCLM4-7-18.v1 always reduces precipitation amounts relative to the lateral boundary forcing it

322 receives, regardless of the driving GCM or the period examined. During December, the precipitation reduction in all

323 RCMs becomes more pronounced and reaches values equal to -3 mm/d. In January, only 1 RCM enhances

324 precipitation (~0.5 mm/d) with all the rest displaying precipitation reduction. During February and March, some

325 positive DIFF values re-appear for some simulations. Overall, there is a strong linear relationship between DIFF in

326 1985-2005 and 2065-2095, which further implies that if an RCM is drier than its driving GCM during the historical

327 period, then it will retain this attribute during the future period also. Nonetheless, we highlight that RCMs preserve

328 precipitation change signal generated by the GCMs. Considering that one primary shortcoming of the GCMs over

329 SAF is their wet bias and that RCMs systematically reduce this bias, we gain increased confidence that RCMs can be

330 reliably used for future projections with regards to precipitation change.

331 In **Fig. 12** the spatial average of the precipitation change signal from RCMs and their driving GCMs relative to 1985-

332 2005 for SAF and the 3 subregions is displayed. Concerning Region A, all models during October identify a

333 precipitation reduction at the end of the 21st century that can reach -0.9 mm/d. The precipitation decline signal is also

334 identified during November, indicating a later onset of the rainy season over SAF, as it has already been shown for





CMIP5 (Dunning et al., 2018). During December and January there is a variability in the spatial averages of the change
signal that ranges from -0.8 to 0.8 mm/d. A similar pattern is also seen for February and March. The distribution of
the ensemble members for both RCMs and GCMs in Regions B and C is similar to that of Region A, however in
Regions B and C precipitation change values display a considerably larger spread. In Region D the climate change
signal is symmetric around 0 for all months, except March.
The impact the RCMs and GCMs on monthly precipitation for the period 2065-2095 under RCP8.5 is shown in **Fig.**
**13**. Regions A and B show a similar behavior as in the historical period (**Fig. 8**), however, in Region C, precipitation
during March is more strongly dominated by GCMs. The same observation holds also for Region D. In general,
regional processes continue to dominate contributions to variability during Oct-Nov, while large scale features
dominate during Dec-Mar.

## 3 Discussion and conclusions

In this work we investigated  whether it is the RCMs or the driving GCMs that control the monthly precipitation
variability, monthly precipitation biases and the climate change signal over southern Africa and how these
relationships vary from month-to-month through the rainy season. Towards this end, we use an ensemble of 19 RCM
simulations performed in the context of CORDEX-Africa and their driving GCMs. According to the literature
(Munday and Washington, 2018), precipitation in the CMIP5 simulations  is characterized by a systematic wet bias
over southern Africa. In the CORDEX-Africa RCM simulations there is also a persistent wet bias, especially during
the core of the rainy season (DJF), however, it is of smaller magnitude and of smaller spatial extent in the RCMs than
the GCMs. It is found that all RCMs  reduce monthly precipitation compared to their driving GCMs for both historical
(1985-2005) and future period (2065-2095) under RCP8.5.
Over Region B, which encompasses Angola Low (AL) activity, the months with the largest biases are found to be
November and March. November is the month during which there is a transition of the AL from a heat low phase to
a tropical low system, and March indicates the end of the rainy season. Hence, precipitation during the transition
months is challenging for both RCMs and GCMs. Over Region C, representing the wider area of Mozambique, the
bias signal is reversed and after January most of the models display a dry bias. Over South Africa (Region D), the
majority of models display a consistent wet bias for all months of the rainy season. All models (CMIP5 and CORDEX-
Africa) display an intense dry bias in the NE part of SAF, which can be related to the misrepresentation of the moisture
transport entering the region from the Indian Ocean (Munday and Washington, 2018). In general, although RCMs
display an improvement of precipitation biases relative to their driving GCMs, still some bias patterns persist even in
RCMs, calling for a process-based evaluation of specific climatological features such as the formulation of the Angola
Low and the transport of moisture from the NE part of SAF towards central SAF.
More specifically, we found that CCLM4-7-18.v1 produces the smallest bias when the whole of SAF is examined,
however, it displays a systematic dry bias over Region C (greater Mozambique region), hence, CCLM4-7-18.v1
should be used with caution over eastern SAF, especially if it is exploited within drought-related climate services.
Concerning RCA4.v1, we find a very regionally heterogeneous -almost pixelated- spatial pattern for precipitation,



which can be attributed to the sharp topography used (Van Vooren et al., 2019). RCA4.v1, due to the large size of its
ensemble, is optimal for analyzing its behavior under different driving GCMs. In general, we find that RCA4.v1 is
more prone to follow the signal received from the driving GCMs, contrary to what is observed for CCLM4-7-18.v1.
REMO2009.v1 presents a compromise between the behaviors of RCA4.v1 and CCLM4-7-18.v1.
It is highly recommended that when RCM simulations are used for the whole of SAF or a subregion thereof, the spread
and statistical properties of all available RCMs and their driving GCMs should be examined and an ensemble of RCMs
should be employed based on their ability to reproduce key climatic features of the region of interest. Increasing
evidence is provided that not all models are fit for constructing an ensemble mean (or median) for all regions (Her et
al., 2019; Raju and Kumar, 2020; Tebaldi and Knutti, 2007). Lastly, a very important aspect when the calculation and
characterization of biases is discussed for GCMs and RCMs, is that biases are assessed based on a satellite or gauge-
based product, which are often erroneously regarded as "the ground truth" (Harrison et al., 2019; Alexander et al.,
2020). Of course, the climate community is bound to work with the state-of-the-science products that are available,
however, biases and errors in the "observational datasets" should be kept in sight when the bias of climate models is
discussed. In this work we use the CHIRPS precipitation product, as it has been shown to outperform other satellite
precipitation products (Toté et al., 2015; Ayehu et al., 2018; Dinku et al., 2018).

Concerning the climate change signal, there is a strong agreement among all GCMs and RCMs that precipitation
during October will decrease by (-0.1) – (-1) mm/d, a fact which is associated with a projected later onset of the rainy
season, which is further associated by a northward shift of the tropical rain belt (Dunning et al., 2018). For the rest of
the months, the results are variable, indicating the need for a multi-model approach, when climate change impacts are
assessed. A feature that is identified in some GCMs and is transferred to the downscaling RCMs, is a precipitation
increase that extends from the central SAF region towards the southeast. This result is consistent with previous work
that  shows an increase in frequency of landfalling cyclones along the eastern seaboard of SAF (Muthige et al., 2018).
Since tropical cyclones are a particular cause of severe flooding events over the region of Mozambique, there is an
urgent need for planning and mitigation strategies over the region.
Lastly, concerning precipitation variability and whether it is the RCMs or the driving GCMs that dominate monthly
precipitation, we find that, as expected, over the whole of SAF (Region A), October and November are dominated by
RCMs, while during Dec-Mar it is the GCMs that mainly formulate the precipitation climatologies. This is explained
by the fact that after December there is a strong large-scale forcing, which is provided to the RCMs by the lateral
boundary conditions given through the GCMs. The results for the historical period are comparable to that for future
projections.
*Code and data availability*
For the data processing and statistical analysis we used the R Project for Statistical Computing ([https://www.r-](https://www.r-)
[project.org/](project.org/)), the Climate Data Operators (CDO) ([https://code.mpimet.mpg.de/projects/cdo/](https://code.mpimet.mpg.de/projects/cdo/)) and Bash programming
routines. Processing scripts are available via ZENODO under DOI: [https://doi.org/10.5281/zenodo.5569984](https://doi.org/10.5281/zenodo.5569984). CMIP5
and CORDEX-Africa precipitation data were retrieved from the Earth System Grid Federation (ESGF) portal



(https://esgf-data.dkrz.de/projects/esgf-dkrz/). The Climate Hazards Group InfraRed Precipitation with Station data
(CHIRPS) products were retrieved from: https://www.chc.ucsb.edu/data/chirps.

*Supplement*
The supplement related to this article is available online.

*Author contribution*
MCK, SPS and EK designed the research. MCK performed the analysis and prepared the manuscript. SPS, EK, LS
and GN edited the manuscript and provided corrections.

*Competing interests*
The authors declare that they have no competing interests.

*Acknowledgements*
This article is funded by the AfriCultuReS project "Enhancing Food Security in African Agricultural Systems with
the Support of Remote Sensing", (European Union's Horizon 2020 Research and Innovation Framework Programme
under grant agreement No. 774652). The authors would like to thank the Scientific Support Centre of the Aristotle
University of Thessaloniki (Greece) for providing computational/storage infrastructure and technical support. MCK
was funded by the Hellenic Foundation for Research & Innovations, under the 2nd Call for PhD Candidates (application
No. 1323).















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



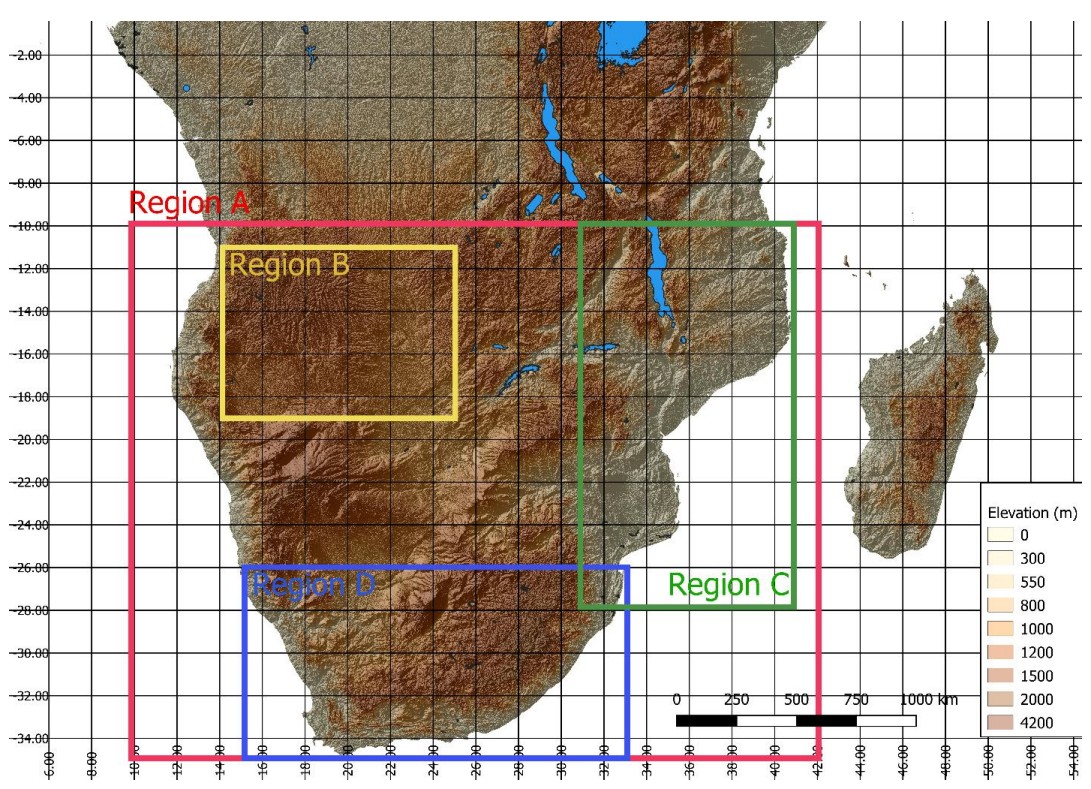

**Figure 1.** Study region and subregions over southern Africa.



**Figure 2.** Monthly precipitation climatologies (mm/d) during October for the period 1985-2005. First column (from the left) displays precipitation from the driving GCMs and columns 2-4 display the downscaled precipitation output from RCA4.v1, CCLM4-8-17.v1 and REMO2009.v1.





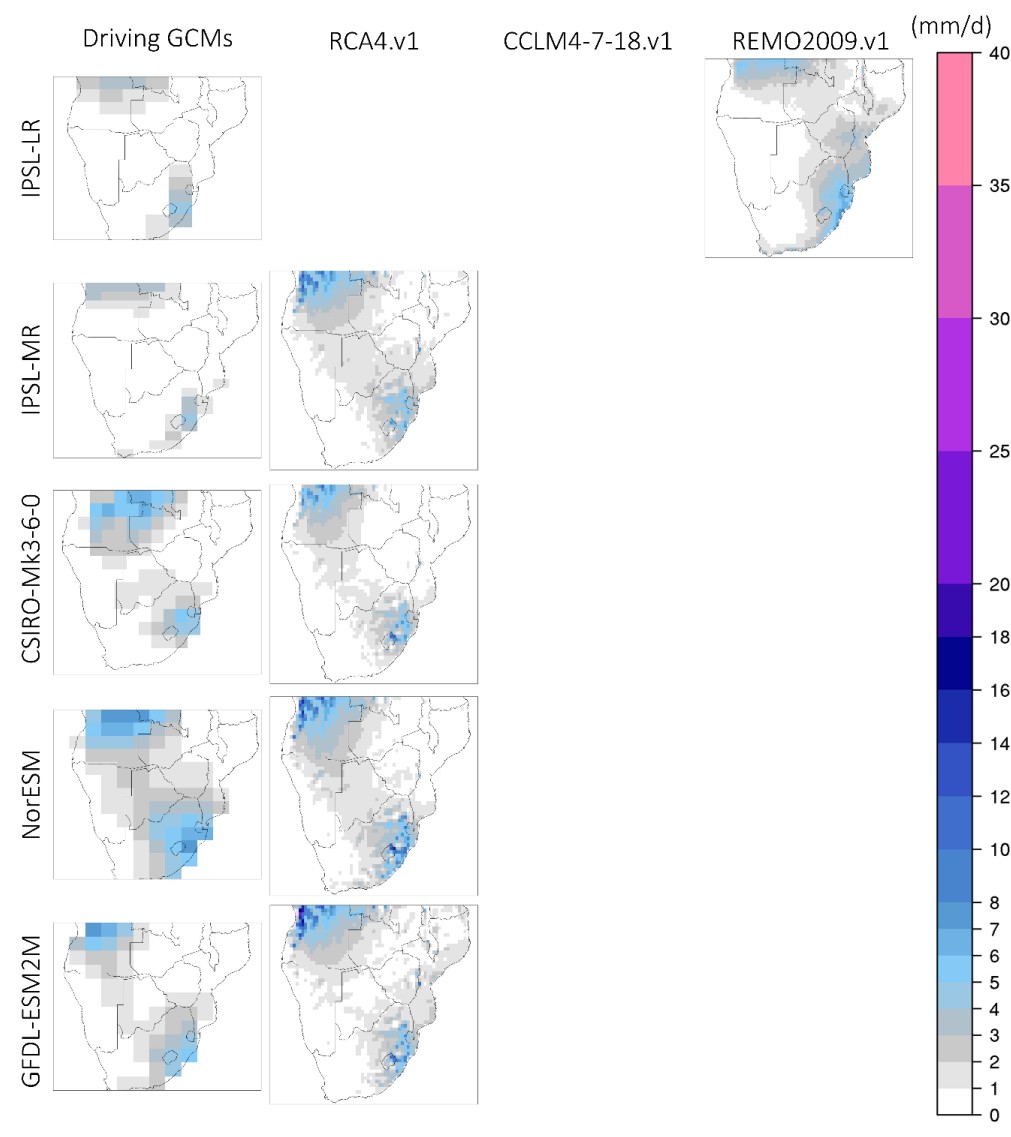


**Figure 2.** Continued.










**Figure 3.** Monthly precipitation climatologies (mm/d) during January for the period 1985-2005. First column (from the left) displays precipitation from the driving GCMs and columns 2-4 display the downscaled precipitation output from RCA4.v1, CCLM4-8-17.v1 and REMO2009.v1.

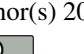



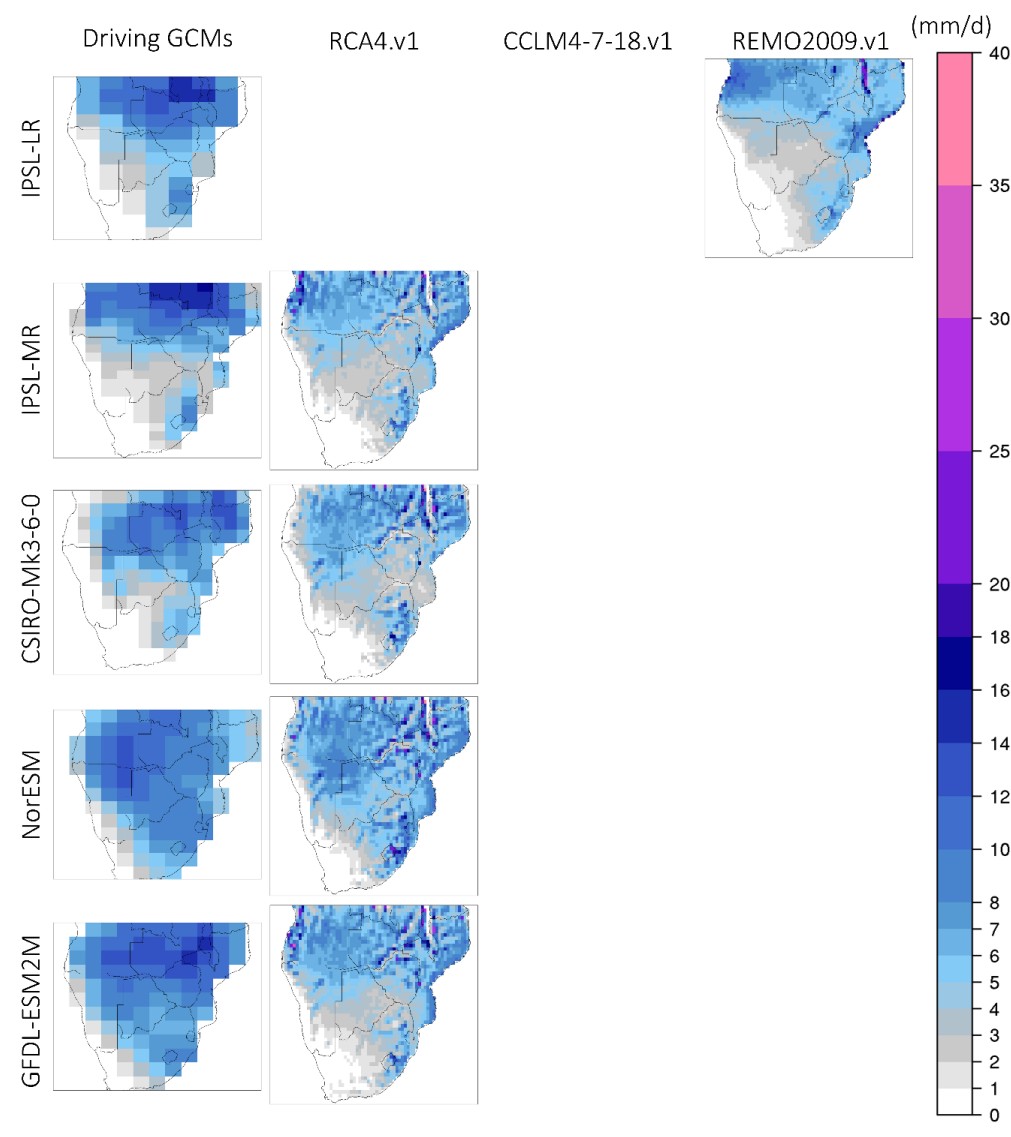


**Figure 3.** Continued.








**Figure 4.** Monthly precipitation bias (model – CHIRPS in mm/d) during October for the period 1985-2005. First column (from the left) displays the biases in the driving GCMs and columns 2-4 display the biases in the downscaled precipitation output according to RCA4.v1, CCLM4-8-17.v1 and REMO2009.v1.



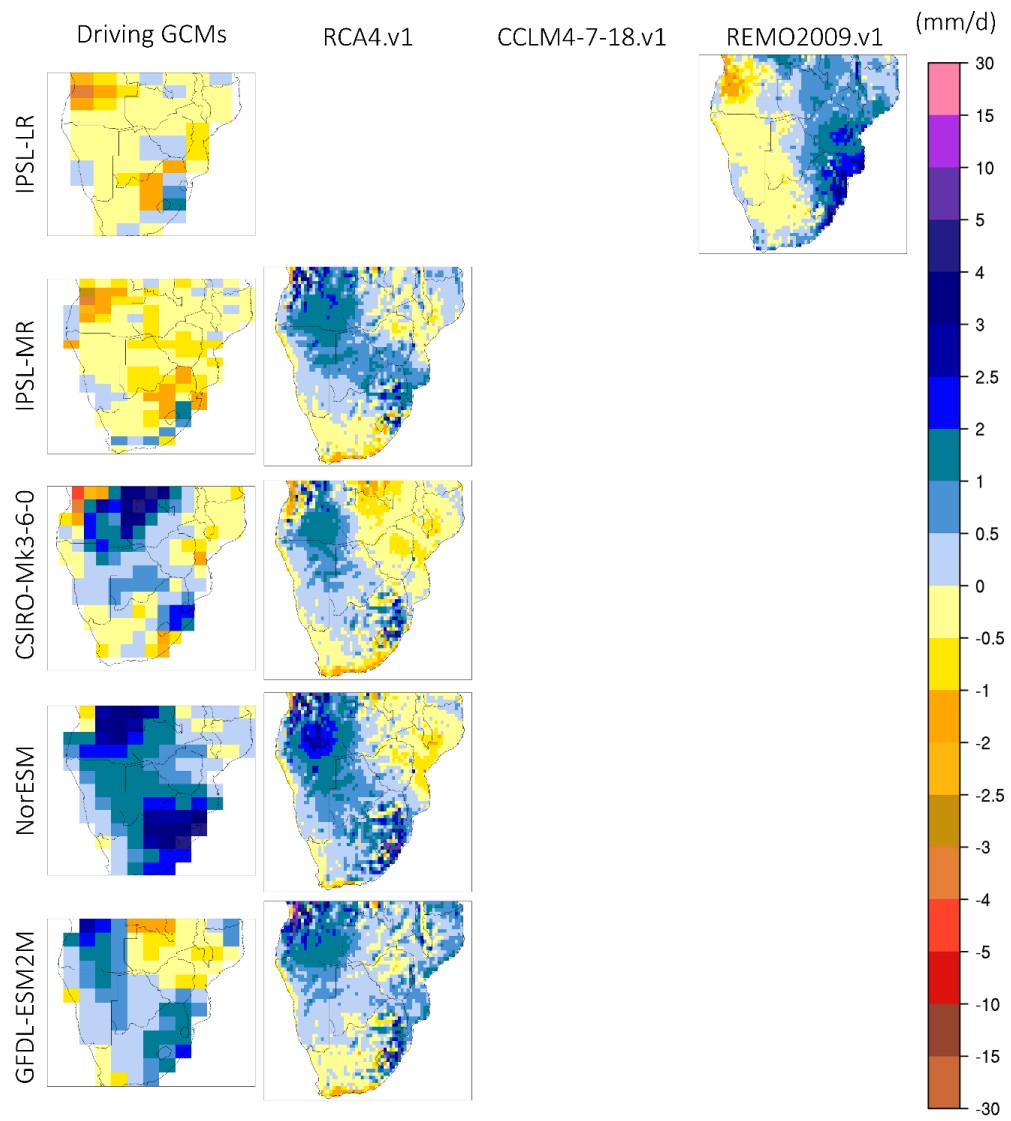


**Figure 4.** Continued.









**Figure 5.** Monthly precipitation biases (model – CHIRPS in mm/d) during January for the period 1985-2005. First column (from the left) displays precipitation biases from the driving GCMs used and columns 2-4 display the downscaled products according to RCA4.v1, CCLM4-8-17.v1 and REMO2009.v1.


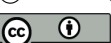



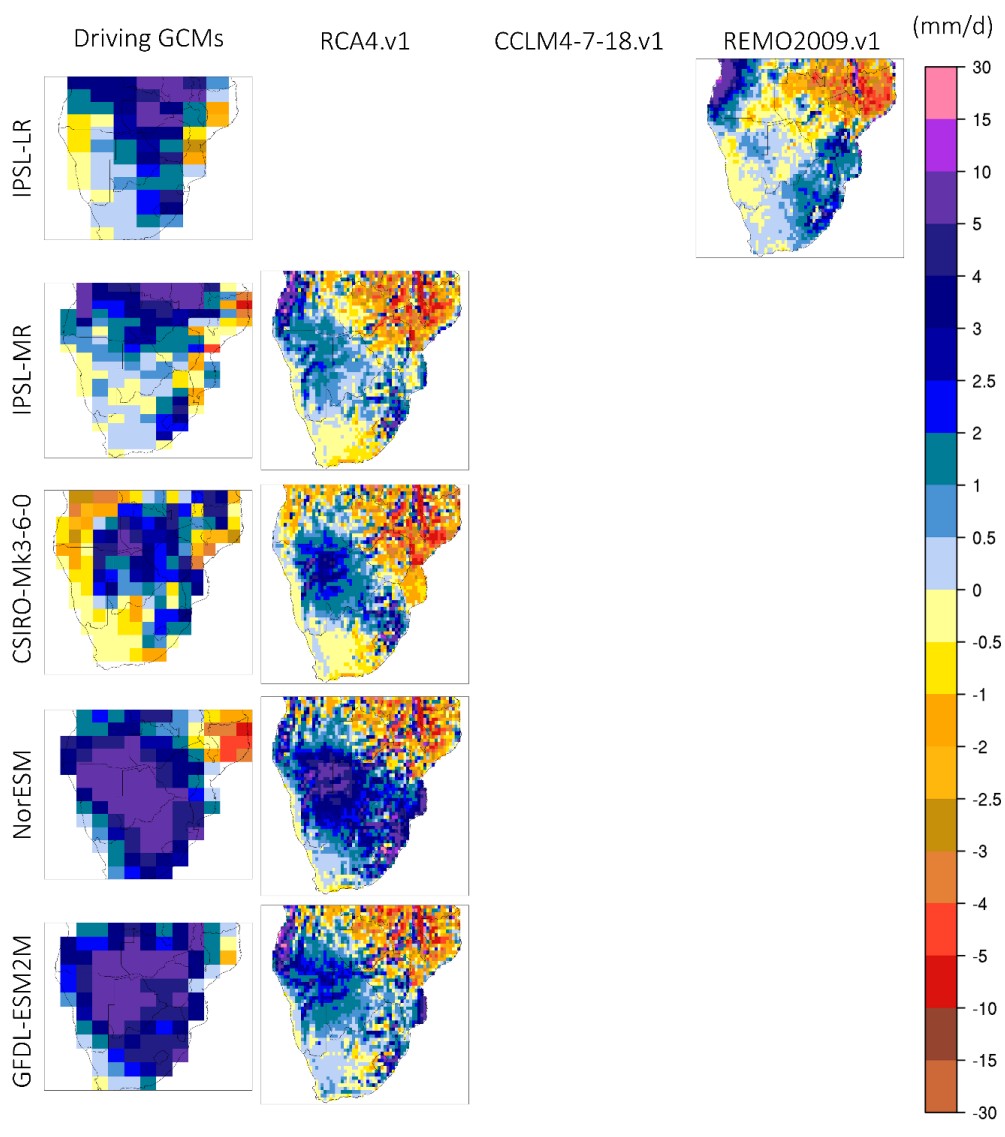


**Figure 5.** Continued.






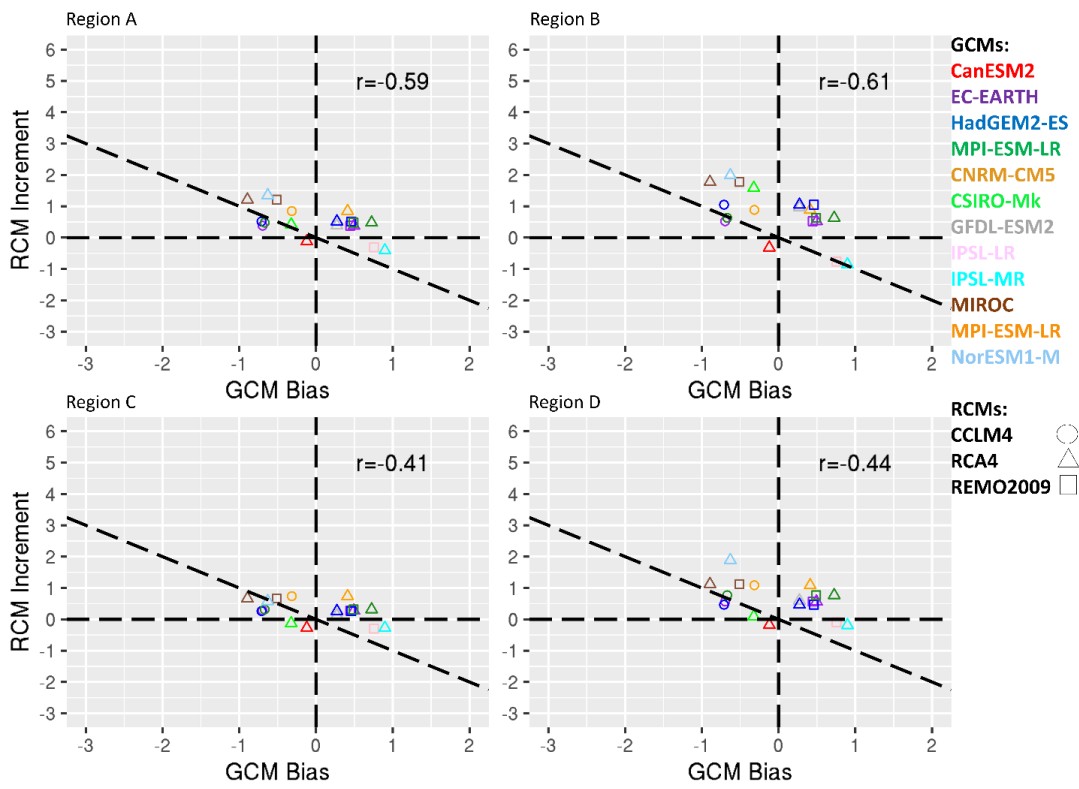

**Figure 6.** Scatterplots of the RCM increment (RCM-GCM) for precipitation (mm/day) as a function of the GCM bias (GCM-OBS) for October. Colors indicate the driving GCM and shapes indicate the downscaling RCMs. The four panels indicate spatial averages over southern Africa (Region A), the Angola Low region (Region B), the Mozambique region (Region C) and South Africa region (Region D).



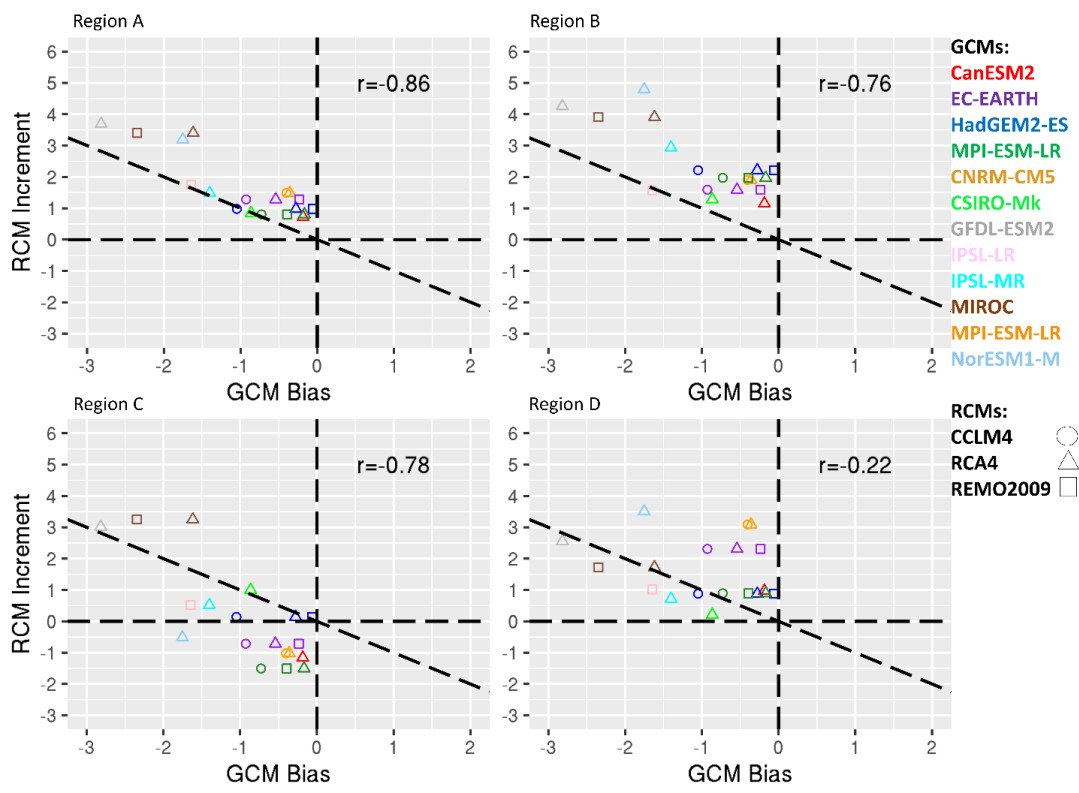

709

**Figure 7.** Scatterplots of the RCM increment (RCM-GCM) for precipitation (mm/day) as a function of the GCM bias
(GCM-OBS) for January. Colors indicate the driving GCM and shapes indicate the downscaling RCMs. The four
panels indicate spatial averages over southern Africa (Region A), the Angola Low region (Region B), the Mozambique
region (Region C) and South Africa region (Region D).






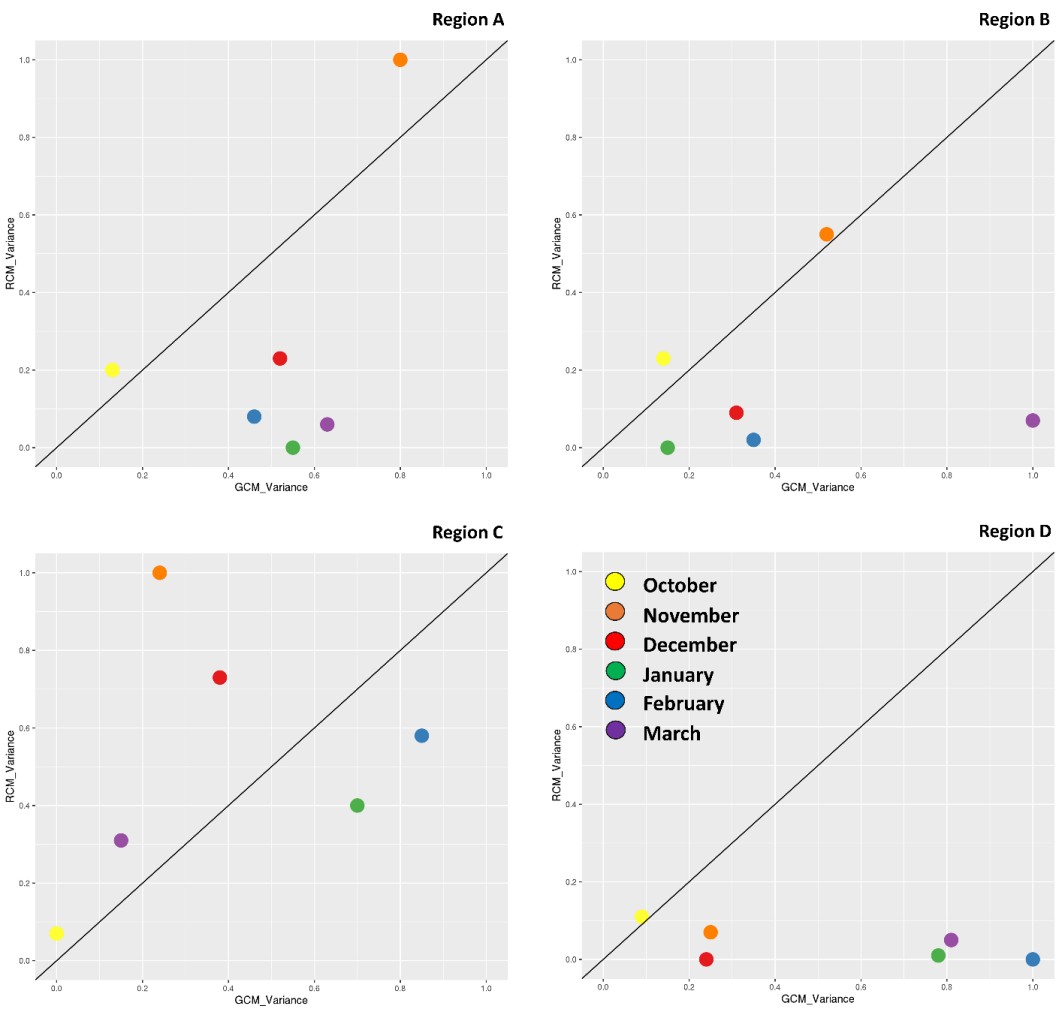

**Figure 8.** Analysis of variance for monthly precipitation during 1985-2005 for southern Africa (Region A) and the 3 sub-regions examined, namely Region B (Angola region), Region C (Mozambique region) and Region D (South Africa region). The x and y-axis display standardized precipitation variances.











**Figure 9.** Monthly precipitation change (future – present in mm/d) during October for the period 2065-2095 relative to 1985-2005. First column (from the left) displays precipitation change from the driving GCMs used and columns 2-4 display the downscaled products according to RCA4.v1, CCLM4-8-17.v1 and REMO2009.v1.





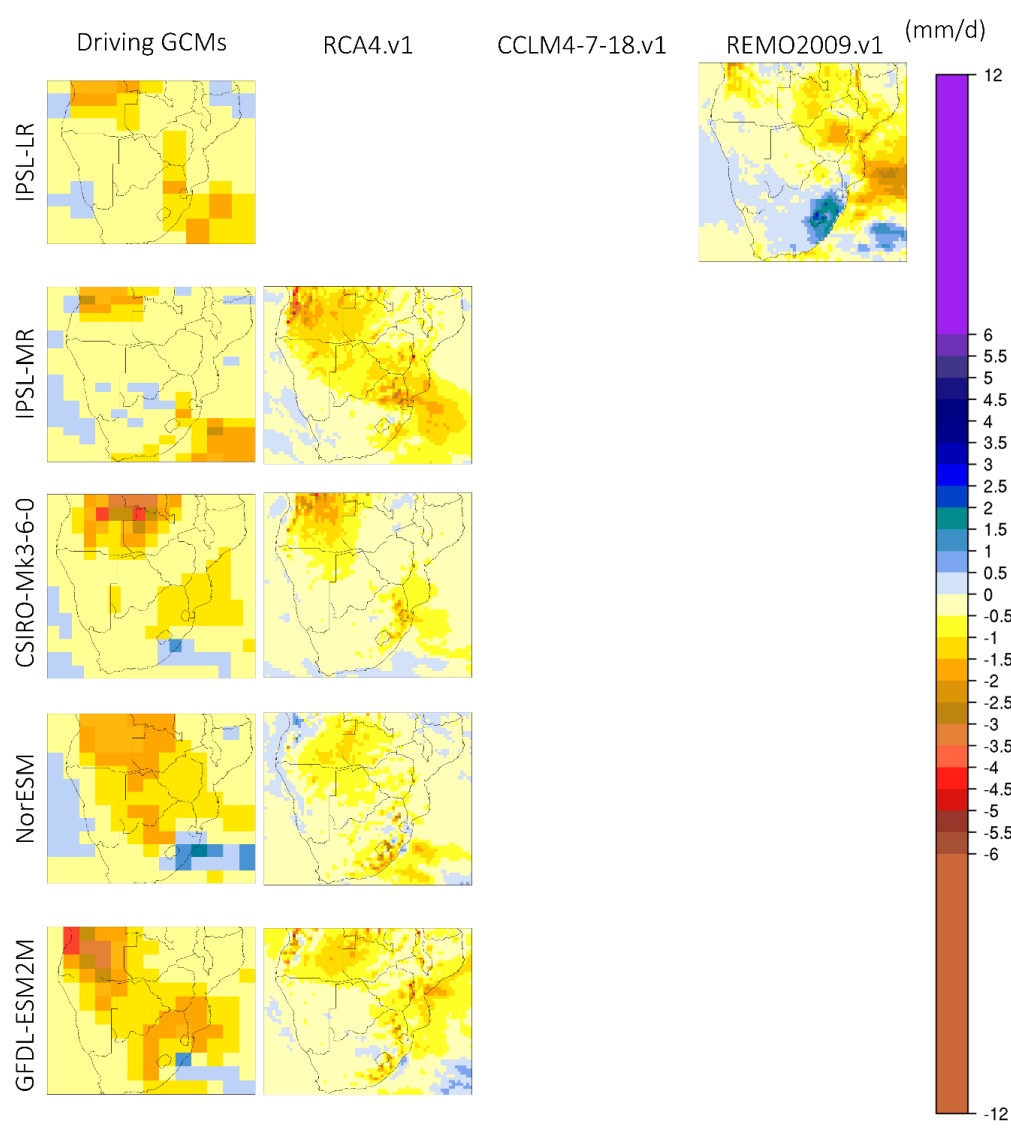


**Figure 9.** Continued.








**Figure 10.** Monthly precipitation change (future – present in mm/d) during January for the period 2065-2095 relative to 1985-2005. First column (from the left) displays precipitation change from the driving GCMs used and columns 2-4 display the downscaled products according to RCA4.v1, CCLM4-8-17.v1 and REMO2009.v1.



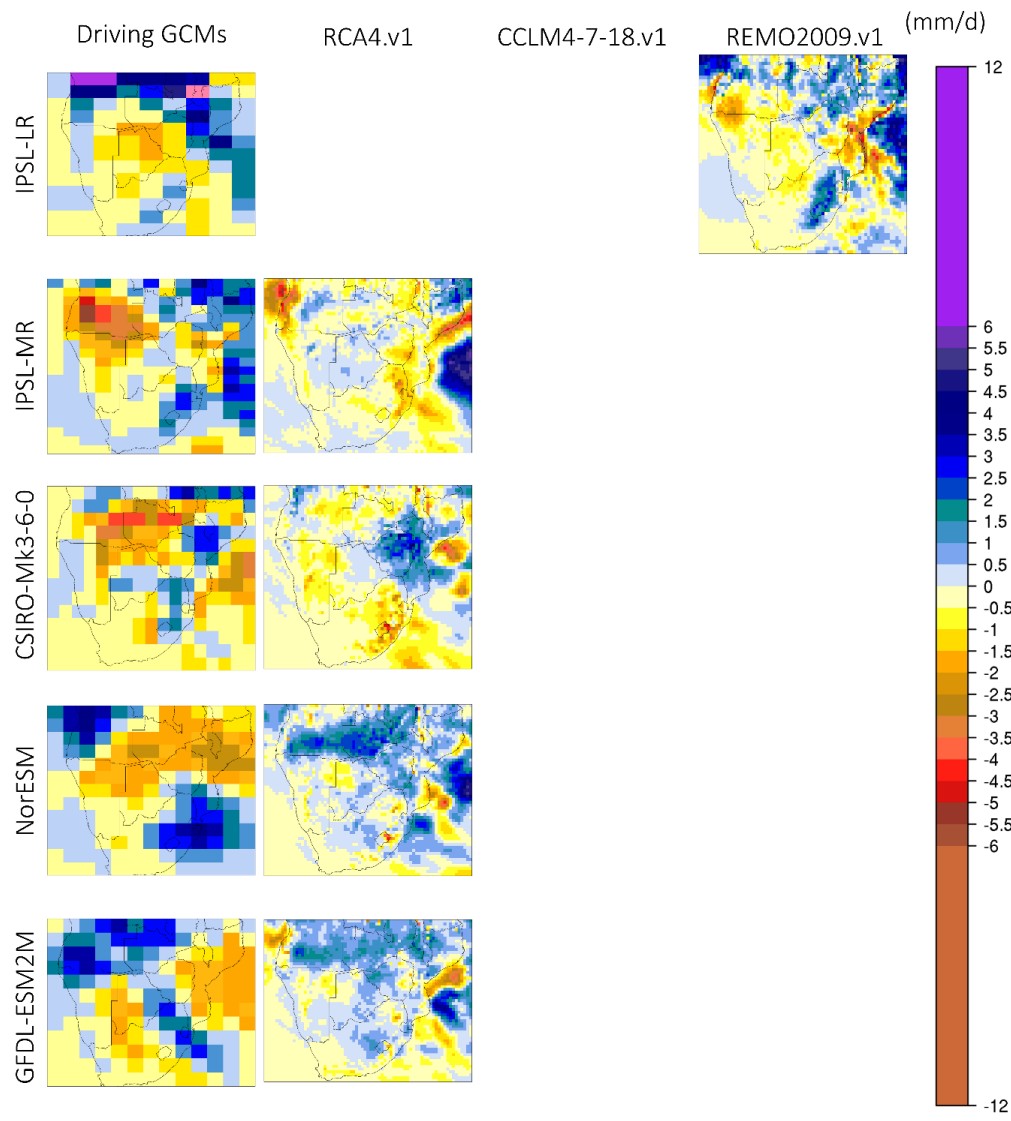


**Figure 10.** Continued.








**Figure 11.** Monthly RCM$_{DRI}$ – DRI spatial averages over southern Africa for the historical period (1985-2005) on the
x-axis and the future period (2065-2095) under RCP8.5 on the y-axis.



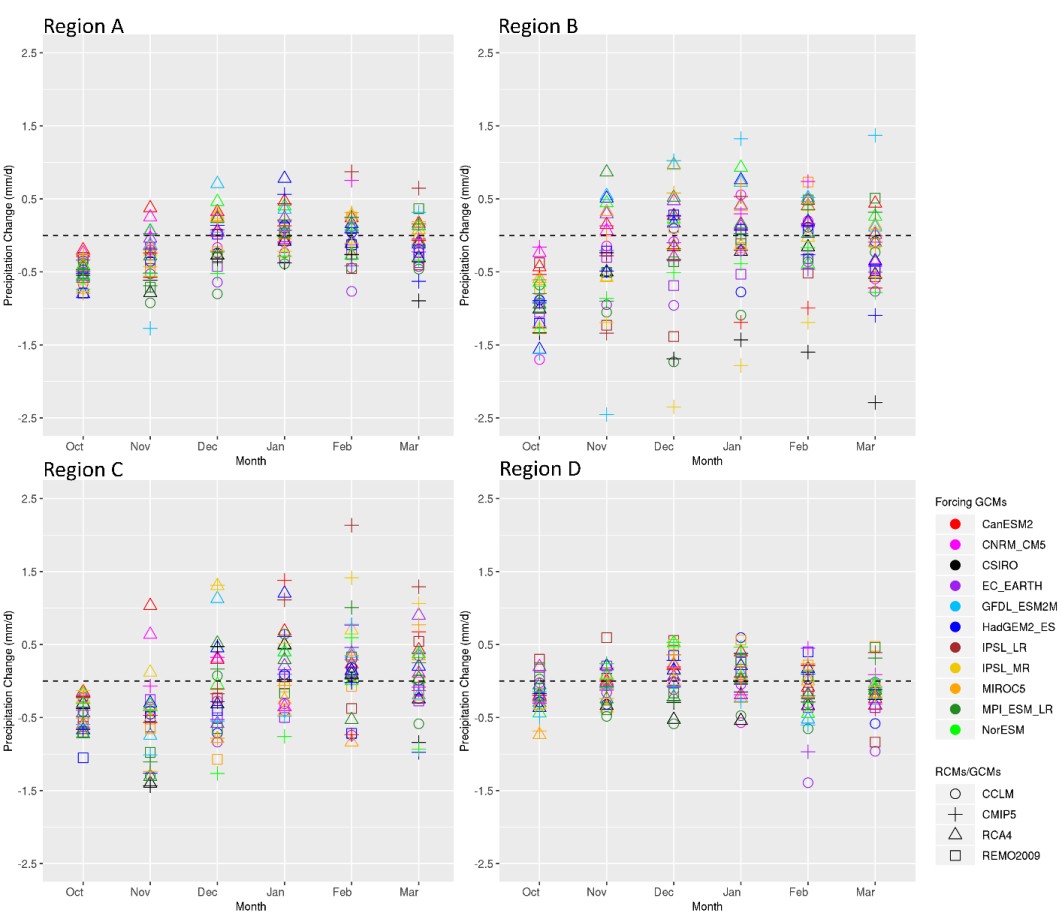


**Figure 12.** Spatial average of the precipitation change signal (mm/d) from RCMs and their driving GCMs relative to
1985-2005 for southern Africa and the 3 sub-regions examined.





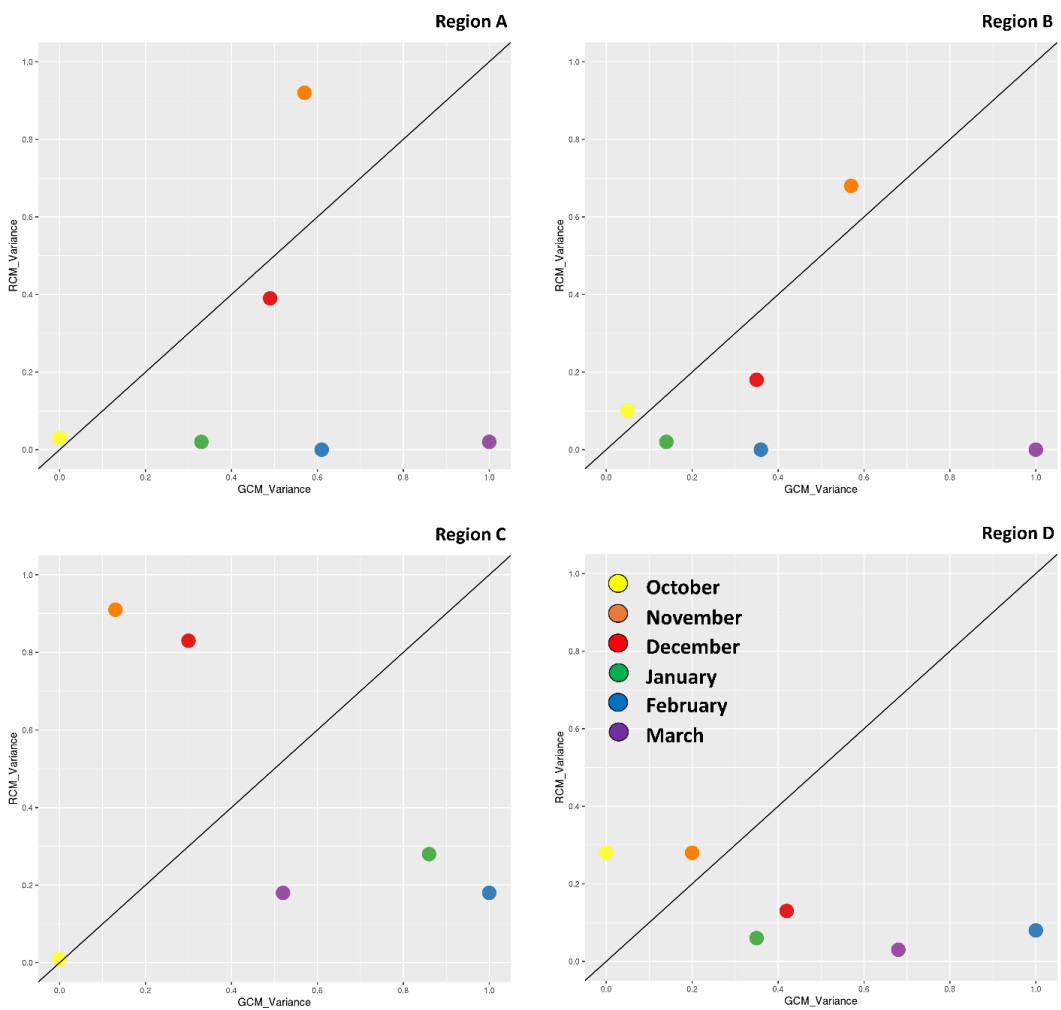

**Figure 13.** Analysis of variance for monthly precipitation during 2065-2095 for southern Africa (Region A) and the 3 sub-regions examined, namely Region B (Angola region), Region C (Mozambique region) and Region D (South Africa region). The x and y-axis display standardized precipitation variances.