# Peer review of "The impact of lateral boundary forcing in the CORDEX-Africa"

_Geoscientific Model Development, 2021_

## Author Comment (AC1)

**Responses to Anonymous Reviewer 2**

**General Comment:**

This is an interesting paper assessing the impact of the GCM as compared to the RCM on a domain in southern Africa. However, I think there are some concerns and questions that need to be addressed before the paper is ready for publication.

Variance between the models (RCMs or GCMs) has been employed to compare the model performance for the simulation of the monthly precipitation. In my reading, this study aims to show how much each RCM output is affected by its driving GCM. However, some concerns that may be critical should be addressed if the information provided in this study will eventually be used by both climate and non-climate scientists, as the authors mentioned.

**RESPONSE**: We would like to thank the Anonymous Reviewer #2 for the positive interpretation of the manuscript. Based on the suggestions and comments, we provide the following replies.

**1st Comment:**

RCM output can be affected not only by the input GCM datasets but also by its parameterization, configuration options, and setup. A simple comparison of the variance of monthly data between the complex models does not guarantee whether the impact of the driving GCM is dominant. All possible outcomes or limitations should be discussed to show the relationships between the RCMs and GCMs.

RESPONSE: Thank you for this comment. Indeed, in a multi-RCM/multi-GCM set of simulations there are multiple sources of uncertainty that may eventually strongly influence the climatological output of a single variable, let alone of a specific atmospheric process studied in a climate-scale context (such as the climatological expression of the Angola Low pressure system). Therefore, monthly variance of a specific variable examined in a multi-RCM ensemble, cannot be attributed solely to the different GCMs that provide the lateral boundary conditions. By no means do we imply that the comparison of variances for monthly precipitation between the driving GCMs and the respective RCMs guarantees that the impact of the GCM is dominant (or not) in all possible aspects. In the revised version of the manuscript, we make clearer statements concerning the limitations and underlying assumptions of our work, however, it would be impossible to technically discuss all possible outcomes or limitations. For example, assessing the impact of the cumulus parameterization scheme on all RCM simulations, would require that all RCMs participating in the CORDEX-Africa ensemble, perform a series of sensitivity runs with -at-least-two cumulus parameterization schemes and make the model output available to the climate modeling community on a database such as ESGF or the Climate Data Sore. Unfortunately, this is not the case. Each research group participating with model runs to the CORDEX-Africa ensemble makes a series of modeling choices that may or may not introduce considerable amounts of uncertainty in the final model simulations. These kinds of studies are performed within specific Flagship Pilot Studies (FPS[1]), which are beyond the scope of the current work. We agree however, that the way through which we frame the argument about the impact of the driving GCMs on RCM simulations may cause some misunderstanding. So, we make clearer statements about some main limitations of our work. These statements are made in the last paragraph of the *Discussion* section and are the following:

*"Lastly, it is imperative to highlight that the impact of the lateral boundary conditions on RCM simulations comprise only a portion of the potential sources of uncertainty in the CORDEX-Africa ensemble examined, therefore attributing entirely the variance of RCM simulations to the driving GCMs would be erroneous. Therefore, we mention that uncertainty in RCM simulations can have a plethora of sources that are mainly categorized as parameter or structural uncertainty (Günther et al., 2020; Howland et al., 2022). These types of uncertainty sources may relate to the parameterization schemes employed by each RCM or assumptions and numerical choices involved in the dynamics of each specific RCM. However, since within CORDEX-Africa only a limited number of variables is being made available to the community, it would be impossible to meticulously*
* * *
[1] https://cordex.org/experiment-guidelines/flagship-pilot-studies/endorsed-cordex-flagship-pilote-studies/

*comment on all possible sources of uncertainty and access the impact of their variance on monthly precipitation."*

In addition, the following sentence is now introduced in the first paragraph of the Discussion section:

*"Our work examines monthly precipitation variance caused by the lateral boundary conditions and does not examine parameter and structural uncertainty separately in the multi-RCM and the multi-GCM ensembles analyzed."*

**2nd Comment:**

How the relaxation zone has been defined to simulate each RCM should be addressed. Bias can be amplified through the lateral boundary conditions.

RESPONSE: Thank you very much for this comment. We have now included this information for all RCM used.

CCLM4-8-17.v1 details are provided in Dosio and Panitz, (2016), according to which 10 grid points are used on every side of the domain. Relaxation is performed using the Davies scheme (Davies, 1976, 1983).

RCA4 solution is relaxed towards the forcing boundary data across an eight-point wide relaxation zone following the Davies' boundary formulation method (Davies, 1976), with a cosine-based relaxation function.

REMO2009.v1 also uses (Jacob et al., 2012) an eight-point wide relaxation zone, following the Davies' boundary formulation method also (Davies, 1976).

Since the relaxation zone used in all three RCMs employs the same method and since eight (RCA4 and REMO2009.v1) and 10 grid points (CCLM4-8-17.v1) are used, it can be assumed that the relaxation zone is not introducing a substantial bias capable of affecting monthly precipitation climatology, trends, and climate change signal, that varies among the three RCMs.

The following portion has now been added to the *Data* section:

*"All RCMs employed a relaxation zone which was either 10 grid-points wide (CCLM4-8-17.v1) or eight points wide (RCA4.v1 and REMO2009.v1). Relaxation in all RCM simulations was performed using Davie's method (Davies, 1976, 1983)."*

**3rd Comment:**

The reasons why the three RCMs have been chosen should be addressed in the data section. If they have shown good performance in the domain, I suggest adding related references.

RESPONSE: The reason for including these three RCMs was not related to their performance over southern Africa, but to the calculation of variances, as stated within the manuscript in the *Data* section: "*More specifically, the CORDEX-Africa simulations selected are those that were driven by more than two GCMs…*".

More specifically, in order to examine the intra-GCM and intra-RCM variance we select all RCM simulations that are driven by at least two (2) GCMs, using the sample variance (n-1), as indicating in the following equation:

$$s^2 = \frac{1}{n-1} \sum_{i=1}^{n} (x_i - \bar{x})^2$$

The sample size required for an acceptable calculation of variance is a non-trivial statistical issue (e. g. https://stats.stackexchange.com/questions/7004/calculating-required-sample-size-precision-of-variance-estimate). Even using three (3) RCM simulations (three simulations of the same RCM receiving lateral boundary conditions from three different GCMs) can be problematic, however, using the "at least three driving GCMs" rule provides an acceptable compromise for the purposes of this analysis.

Based on our search in the ESGF-DKRZ database (https://esgf-data.dkrz.de/search/cordex-dkrz/), the RCM simulations that are driven with more than two (2) GCMs are: CCLM4-8-17, RCA4, REMO2009, as displayed in the figure below (last accessed at 25/7/2022).

The RACMO22T simulations were performed using lateral boundary conditions from HadGEM2-ES and EC-EARTH (r1i1p1 and r12i1p1) for the historical simulations and from HadGEM2-ES and EC-EARTH (r1i1p1) for the RCP8.5 simulations, hence it did not meet the >2 GCMs criterion and was therefore excluded from the analysis.

[Figure]

*Figure 1: ESGF-DKRZ database search at 25/7/2022.*

The statement in the *Data* section is now changed from:

"More specifically, the CORDEX-Africa simulations selected are those that were driven by more than two GCMs and for which there are runs available for both the historical and the future period under RCP8.5."

to:

"More specifically, the CORDEX-Africa simulations selected are those that were driven by more than two GCMs **(at least three simulations available using the same RCM driven by at least three different GCMs)** and for which there are runs available for both the historical and the future period under RCP8.5."

**4th Comment:**

l57 - The authors should perhaps also cite the RCM lateral boundary papers focusing on southern Africa authored by Ditiro Moalafhi, given the focus there is the impact of changes to lateral boundary conditions, with bias assessed based on a reanalysis dataset. These papers are:

1   Moalafhi, D. B., Sharma, A., Evans, J. P., Mehrotra, R. & Rocheta, E. Impact of bias-corrected reanalysis-derived lateral boundary conditions on WRF simulations. Journal of Advances in Modeling                                    Earth                                    Systems                                    (2017).

2   Moalafhi, D. B., Sharma, A. & Evans, J. P. Reconstructing hydro-climatological data using dynamical downscaling of reanalysis products in data-sparse regions–Application to the Limpopo catchment in southern Africa. Journal of Hydrology: Regional Studies 12, 378-395 (2017).

3   Moalafhi, D. B., Evans, J. P. & Sharma, A. Influence of reanalysis datasets on dynamically downscaling          the          recent          past.          Climate          Dynamics          49,          1239-1255          (2017).

4   Moalafhi, D. B., Evans, J. P. & Sharma, A. Evaluating global reanalysis datasets for provision of boundary conditions in regional climate modelling. Climate Dynamics 47, 2727-2745, doi:10.1007/s00382-016-2994-x (2016).

**RESPONSE**: We thank Reviewer 2 for the suggested papers, which we have read. All four (4) papers provide interesting aspects to the discussion of the impact of the lateral boundary conditions on regional climate modeling, however, they are relevant to primarily model-specific issues and not to the broader discussion developed within the CORDEX community, with regards to the added-value of regional climate modelling (relative to the driving GCMs) and to how the RCM-GCM simulation matrix could be optimized, in order to advance the understanding of the research community on basic-science climatological issues and to also provide reliable tools for climate-services.

More specifically, in Moalafhi et al., 2017a (1), it is stated that there are "… *inconsistencies between the impact of the bias correction prior to downscaling and the resultant model simulations after downscaling. Mean and standard deviation bias-corrected WRF simulations are, however, found to be marginally better than mean only bias-corrected WRF simulations and raw ERA-I reanalysis-driven WRF simulations. Performances, however, differ when assessing different attributes in the downscaled field. This raises questions about the efficacy of the correction procedures adopted.*"

This statement summarizes the challenges posed by the selection of lateral boundary conditions on regional model simulations and also, the sensitivity of the assessment process to the specific atmospheric variable that is assessed. In such variable-specific assessments, a specific variable may display improved performance (prior and before to dynamical or statistical downscaling), however this may happen in isolation to the rest atmospheric variables. In brief, it is common that certain model-specific choices yield the right answer for the wrong reason. There are not absolute methodological choices on how to avoid such modeling "traps", however, a good practice is to

perform a process-based assessment. In our work, we use the selected subregions and the specific months of the rainy  season, as a proxy of the attributes of specific region-specific and month-specific climatic features. In such a way, we aim on performing an indirect process-based assessment of the impact of the lateral boundary conditions on RCM simulations. Therefore, we think that our work and the work of Moalafhi et al., 2017a (1) has fundamental differences on the methods applied and, on the conclusions drawn.

Moreover, Moalafhi et al., 2017b (2) is a model evaluation study in which the ERA-Interim reanalysis dataset is dynamically downscaled using WRF, with an emphasis on hydrological applications over the Limpopo catchment. Although the conclusions having applications for hydrological modeling are interesting, the dynamical downscaling of a reanalysis dataset such as ERA-Interim, is a standard procedure in regional climate modeling. This work addresses different research questions compared to those set in the current paper under review.

Furthermore, in Moalafhi et al., 2017c (3) WRF is driven by two different reanalysis products, ERA-Interim and MERRA. Although the context of Moalafhi et al., 2017c (3) is different to that described in the current paper under review, still its main conclusion on the impact of the lateral boundary conditions on the RCM simulations is in accordance with the context in which the current work is performed. For this reason, Moalafhi et al., 2017c (3) is used as a citation in Line 72.

The sentence from:

"Still, uncertainty arising from both the driving GCM and the downscaling RCM affect the final product… "

Is now changed to:

"Still, uncertainty arising from both the driving GCM (Moalafhi et al., 2017) and the downscaling RCM affect the final product (Nikulin et al., 2012)… "

Lastly, in Moalafhi et al., 2016 (4), five reanalysis datasets are evaluated with the purpose to identify the optimal lateral boundary conditions dataset. The work of Moalafhi et al., 2016 addresses very different research questions compared to those set in the current paper under review.

**5th Comment:**

l89 - I would like to point the authors to papers on correcting lateral and lower boundary variables focusing on Australia, where the focus was the representation of drought, and the RCM used was WRF. While here the authors are using an ensemble of GCMs, these papers focused more on what the representation of different attributes in the lateral and lower boundaries did to the overall monthly outcomes. These papers are:

1    Rocheta, E., Evans, J. P. & Sharma, A. Correcting lateral boundary biases in regional climate modelling: the effect of the relaxation zone. Climate Dynamics 55, 2511-2521, doi:10.1007/s00382-020-05393-1                                                                                                      (2020).

2    Rocheta, E., Evans, J. P. & Sharma, A. Can bias correction of regional climate model lateral boundary conditions improve low-frequency rainfall variability? Journal of Climate 0, null, doi:10.1175/jcli-d-16-0654.1 (2017).

Of special interest to this study is the first paper that assessed the progressing deterioration in the corrections as one focused deeper into the domain (away from the relaxation zone). I think the authors have missed this entire volume of work as I see no references to these papers. Please also note the different levels of impact lower versus lateral boundaries end up having on the simulations.

**RESPONSE**: Line 89 (and 88) to which this comment refers states the following:
> "*In this work we aim to assess whether it is the RCMs or their driving GCMs that dominate monthly precipitation climatology, monthly precipitation bias and climate change signal over SAF.*" We do not claim that GCMs control entirely monthly precipitation climatologies, biases and climate change signal, but rather, we aim to identify the dominant agent (between RCMs and driving GCMs).
>
> Both papers suggested above, involve technical methodological details about bias correcting the lateral boundary conditions provided by GCMs as input to the RCM simulations. The work currently under review exploits the dynamically downscaled RCM simulations performed within CORDEX-Africa. None of the RCMs analyzed here employs statistical downscaling methods or a prior bias correction of the lateral boundary conditions providing information to the RCMs.
>
> In addition, the papers proposed for citation above, refer to a different study region, with a different morphology and coastline, which is affected by entirely different large scale circulation patterns, than southern Africa. The impact of the domain set-up and size is very different between the two study regions and therefore we think that the results drawn for Australia are not transferable to the region of southern Africa.

**6th Comment:**

I also urge the authors to read the additional more recent papers:

1  Kim, Y., Evans, J. P., Sharma, A. & Rocheta, E. Spatial, temporal, and multivariate bias in regional climate model simulations. Geophysical Research Letters 48, e2020GL092058 (2021).

2  Kim, Y., Rocheta, E., Evans, J. P. & Sharma, A. Impact of bias correction of regional climate model boundary conditions on the simulation of precipitation extremes. Climate Dynamics 55, 3507-3526, doi:10.1007/s00382-020-05462-5 (2020).

Here the focus was on extremes, which were found to be impacted to a greater extent by the lateral boundary corrections, than the overall monthly attributes. Please also note the more recent of the two papers that attempted to quantify the impact of multivariate dependence bias in the lateral and lower boundaries, noting that the lack of this plays a significant role in the over quality of simulations.

I must confess that I am an author to the above papers and leave it to the authors (and editor's) judgement whether my suggestions above are essential to the present study. However, in my reading of the current paper, I did feel the above referenced works do add to the story the authors are trying to tell, as they focus on a similar domain (Moalafhi) and altered lateral boundaries (all).

RESPONSE: The 2nd paper suggested (Kim et al., 2020) has now been cited in the *Introduction* section (4th paragraph).

This sentence:

"*For instance, when there is a strong large-scale circulation signal that is introduced to an RCM domain (e.g. advective mid-latitude storms), it is quite likely that the RCM will be able to reproduce the information that is received at its lateral boundaries.*"

Has now been changed to:

"*For instance, when there is a strong large-scale circulation signal that is introduced to an RCM domain (e.g. advective mid-latitude storms), it is quite likely that the RCM will be able to reproduce the information that is received at its lateral boundaries,* **however, the GCM's impact on the RCM  simulation may also vary depending on how far a region lies from the RCM domain boundaries (Kim et al., 2020).**"

**7th Comment:**

l172 - Please clarify if this is for the mean or the monthly series.

RESPONSE: In Lines 168-170 the following is mentioned:

*"As an exploratory method of inspecting the differences between each RCM simulation from its respective driving (GCM) for **monthly precipitation** during both the historical and the future period, we subtract the downscaled precipitation field ($RCM_{DRI}$) from its driving (DRI), as in **Eq. 1**:"*

In line 172 to which this comment refers we have now added the following:
"If *DIFF>0* **(monthly precipitation)**..."

**8th Comment:**

l184 - There seems to be a mistake in the notation here, or also in equation 2. Please check. GCMi should refer to the variance of all RCMs driven by GCMi? Also, what is N?

RESPONSE: N refers to the number of available simulations contributing to either the inter-RCM or inter-GCM variance

The following changes have been made to the equations, to make notation clearer:

From:

$$RCM_{var} = \frac{1}{N_{RCM}} \Sigma_{RCM_j} \left( P_{ij} - \underline{P_j} \right)^2 \qquad \textbf{Eq. 2}$$

To:

$$RCM_{var} = \frac{1}{N_{RCM}} \Sigma_{RCM_j} \left( P_j - \overline{P_j} \right)^2 \qquad \textbf{Eq. 2}$$

From:

$$Inter\_RCMvar = \frac{\Sigma GCM_I}{N} \qquad \textbf{Eq. 3}$$

To:

$$Inter\_RCMvar = \frac{\Sigma GCM_j}{N} \qquad \textbf{Eq. 3}$$

From:

$$GCM_{var} = \frac{1}{N_{GCM}} \Sigma_{GCM_i} \left( P_{ij} - \underline{P_i} \right)^2 \qquad \textbf{Eq. 4}$$

To:

$$GCM_{var} = \frac{1}{N_{GCM}} \Sigma_{GCM_i} \left( P_i - \overline{P_i} \right)^2 \qquad \textbf{Eq. 4}$$

**9th Comment:**

l187 - similar confusion about the notation as earlier - please check and correct.

RESPONSE: This has been addressed in the previous comment. Thank you!

**10th Comment:**

Figure 2 and 3 - It would have been nice to also show the observed climatology in this figure.

RESPONSE: Thank you for this suggestion! We have now added a panel where precipitation climatology for the months of the rainy season (Oct-Mar) is shown for: ERA5, CHIRPS, CRU, UDEL, and MSWEP.

**11th Comment:**

Remaining figures and ANOVA analysis - I found this quite comprehensive and interesting to read. I realize the authors are already reporting a lot of information here, but I was interested on their comments on the following:

(a) What was the impact on temperature simulations and how the intra-RCM variances there compared against the precipitation?

(b) There is little focus on variability, although the main advantage an RCM brings is the added variability in both space and time. Could the authors comment on within grid variabilities across the RCMs and change in variability in time at each grid cell?

(c) Our results showed significant impact on precipitation extremes (from altered lateral and lower boundaries). It would be interesting if the authors could comment on this aspect of the RCM simulations compared to the GCM simulations.

RESPONSE: Thank you for this comment.

(a) In our work, the variable in concern is precipitation and the degree to which our conclusion can be generalized for other variables is a topic of further research. However, similar work has been performed for other study regions within CORDEX:

- Boberg F and Christensen J H 2012 Overestimation of Mediterranean summer temperature projections due to model deficiencies Nat. Clim. Change 2 433–6; -
- Christensen J H and Boberg F 2012 Temperature dependent climate projection deficiencies in CMIP5 models Geophys. Res. Lett. 39 L24705
- Sørland SL, Schär C, Lüthi D, Kjellström E (2018) Bias patterns and climate change signals in GCM-RCM model chains. Environ Res Lett 13:. https://doi.org/10.1088/1748-9326/aacc77).

b) Indeed, RCMs do increase the interannual variability of monthly means compared to their driving GCM. Also, RCMs -due to their higher horizontal resolution- increase the spatial variability of a field, especially precipitation which displays strong spatial heterogeneity. The increased spatial variability in RCMs can also be observed through visual inspection in the maps of monthly precipitation climatology and on maps displaying climate change signal. However, this work was concerned about analyzing climate mean values over a period indicative of the "current climate" and over a future period under RCP8.5. However, we do agree that investigating the modulation of interannual variability from the GCM(i) to the RCM(i) would be an interesting point to further study.

c) We do agree that investigating the degree to which precipitation extremes in RCMs are affected by the driving GCMs is also a very interesting topic. However, considering that precipitation extremes are expected during the core rainy season (DJF) during which

precipitation is mainly caused by large-scale circulation that is significantly affected by the driving GCMs, this would introduce an additional level of complexity and it would perhaps comprise a complete analysis on each own.

---

## Author Comment (AC2)

**Responses to Anonymous Reviewer 1**

**General Comment:**

The impact of lateral boundary forcing in the CORDEX-Africa ensemble over southern Africa by Karypidou et al.

This manuscript attempts to answer the challenging question regards the extent to which RCM simulations can reduce biases present in GCM simulations for the regional climate of southern Africa. The manuscript is well-written with clearly documented methods, a well-justified aim, and appropriate figures.

There are a few minor and major queries or difficulties I had in understanding these results, which I detail here.

RESPONSE: We would like to thank the Anonymous Reviewer #1 for the positive interpretation of the manuscript. Based on the suggestions and comments, we provide the following replies.

Major Comments

**1st Comment:**

Variance analysis presented in Fig.8 and 13.: These results are crucial to the stated aims of the manuscript. However, I am left wondering to what extent the physical interpretation made in this analysis is undermined by the low number of RCM members (three) along with a sensitivity to total rainfall in the following way:

1. Variance is constrained to be lower in October than January. Indeed, much of southern Africa only experiences full onset of rains by November. So, any intermodel variability is constrained by total rainfall. This seems reflected by Fig. 8 C and D, where GCM variance is higher in the respective region's wetter months. Similarly, in October in the future [Fig. 13] all variance is very low, reflecting the lack of rain in this month.

2. The behaviour set out in a. would be seen in a well-sampled system (e.g. 12 GCMs) but with 3 RCMs there the is a high risk that either they all look the same and there is no variance or there is one that is very different and the variance is substantial. This happens here as CCLM4 is biased dry in October with the others two biased wet.

3. The behaviour in b. is strongly dependent on the behaviour of only one RCM. If that RCM comes in line with the others later in the season, as CCLM does with similar wet/dry bias patterns to the other RCMs in January, the variance is much lower. Indeed, in region B and D it is surprisingly almost zero in some months.

4. Part of this query may be unpicked if the rainfall is standardized per month in order to remove problem (a). I'm not sure if this will address b and c though.

RESPONSE: Thank you for all the points raised! We provide the answers below.

> Following Vautard et al. (2021) we apply a common standardization of precipitation variance across all months and all subregions examined. We do that in order to place precipitation variances emanating from RCMs and GCMs on a scale that would range from 0 to 1, and thus provide easily comparable results for all subregions and all months. A schematic of the method we use for calculating inter-RCM and inter-GCM variances is displayed in Figure S1 of Supplementary material. We do agree that the fact that October is much drier than January and the fact that the common standardization is applied for all months, results in variance being constrained for October. However, our main aim through these plots (Figures 8 and 13, now Figures 9 and 14) is to distinguish between RCM or GCM dominated precipitation variances. Our primary goal is not to make a statement about the actual value of standardized precipitation variance, but rather to examine our hypothesis of RCMs dominating precipitation signal in the early rainy season (ON) and GCMs dominating precipitation in the core rainy season (DJF). The actual magnitude of precipitation variance will be affected by the way standardization is performed; however, it would be problematic and therefore challenge our initial hypothesis, had the standardization method affected the classification of each month to GCM or RCM dominated regimes. In fact, we have now performed a standardization in which minimum and maximum precipitation variance are performed for each specific month separately. The results are displayed below. The two main

conclusions are that October and November remain on the upper triangle (RCM dominated regime), while Dec-Mar remain on the lower triangle (GCM dominated regime), as in the original plot contained in the manuscript. Of course, the new standardized values of precipitation variance are changed, since standardization is performed for each month separately, and each month in concern displays the highest variance (1) in each respective triangle. Because standardization using October minimum and maximum values was very low, the rest of the months do not appear in the respective month's panel. Same is the case for November and December.

[Figure]

We also note that standardization of precipitation variances was performed using the following formula:

$$Norm = \frac{x - min(x)}{max(x) - min(x)}$$

$x$: Precipitation variance for each month

$min(x)$: Minimum precipitation variance for all months for all subregions

$max(x)$: Maximum precipitation variance for all months for all subregions

In addition, and with regards to the fact that variance analysis is not well-sampled (only three RCMs participate in the CORDEX-Africa ensemble) we agree that this is far from optimal, however, it is a necessary compromise imposed by data availability. There are statistical workarounds in "filling-the-gaps" in the GCM-RCM simulation matrices, however, they also come with a set of considerable deficiencies (Christensen and Kjellström, 2022).

Christensen, O. B. and Kjellström, E.: Filling the matrix: an ANOVA-based method to emulate regional climate model simulations for equally-weighted properties of ensembles of opportunity, Clim Dyn, 58, 2371–2385, https://doi.org/10.1007/s00382-021-06010-5, 2022.

Vautard, R., Kadygrov, N., Iles, C., Boberg, F., Buonomo, E., Bülow, K., Coppola, E., Corre, L., van Meijgaard, E., Nogherotto, R., Sandstad, M., Schwingshackl, C., Somot, S., Aalbers, E., Christensen, O. B., Ciarlo, J. M., Demory, M.-E., Giorgi, F., Jacob, D., Jones, R. G., Keuler, K., Kjellström, E., Lenderink, G., Levavasseur, G., Nikulin, G., Sillmann, J., Solidoro, C., Sørland, S. L., Steger, C., Teichmann, C., Warrach-Sagi, K., and Wulfmeyer, V.: Evaluation of the Large EURO-CORDEX Regional Climate Model Ensemble, Journal of Geophysical Research: Atmospheres, 126, e2019JD032344, https://doi.org/10.1029/2019JD032344, 2021.

**2ⁿᵈ Comment:**

Based on my understanding of recent literature for the region, I disagree with the interpretations about the regional climate drivers.

1. Munday & Washington 2018 demonstrated the heat low to tropical low switch of the Angola Low. Howard & Washington showed the tropical Angola low was in fact the monthly aggregate of frequent tropical depressions crossing southern African from Mozambique and stalling in Angola. This is at odds with the interpretation provided in line 143-145, where the Angola Low is viewed separately.
2. In the Heat Low phase, the Angola Low is not directly driving rainfall. It cannot because it can only develop under subsiding clear-sky conditions. Rainfall in the early season happens when the heat low is temporarily displaced/dissolved. The leading candidate for this displacement is large-scale synoptic westerly waves. See c.
3. Work from the early 1990s by D'Abreton and picked up by others, including recently Hart et al 2018, suggested southern African rainfall is controlled by mid-latitude westerly wave dynamics (large-scale) earlier in the season. This then gives way to more local processes later in the season as the moist thermodynamic environment becomes more tropical (less subtropical) by the height of summer. Your speculation (Line 25-28) counters this, which is fine, but see d. below.
4. Speculation about land-surface coupling seems key to your argument, but at least as far as I am aware this is not well-established for early season over southern Africa. Please include references which point to this if you have them available. I am not sure if the literature, as yet, has shown that for example the soil-moisture – rainfall coupling seen in Indian, and the Sahel does play a role in southern Africa. And it is an open question whether this is true in the real-world, let alone whether it is resolved in RCMs.

**RESPONSE**: Thank you for this comment and all the issues raised. We acknowledge their importance to the theoretical assumptions of our work and address it point-by-point below.

1. We agree with this statement. We also consider the Angola Low as a climatic feature that switches from the heat low phase at the early rainy season to the climatological aggregate of transient depressions during DJF. The phrasing over lines 143-145 (line numbering prior to revision) is now corrected to the following: "*Since precipitation during Dec-Feb is caused by the tropical low phase of the Angola low pressure system, which is the monthly aggregate of frequent transient low pressure systems crossing southern African (Munday and Washington, 2017; Howard and Washington, 2018; Howard et al., 2019), we hypothesize that the impact of the driving GCM fields during Dec-Feb is enhanced*".

2. We do recognize the importance of large-scale synoptic westerlies; however, the CORDEX-Africa ensemble does not allow a detailed analysis of the properties of the upper-level westerly flow because of the geographical extent of the CORDEX-Africa domain

(https://cordex.org/domains/region-5-africa/). More specifically, the southernmost boundary of the CORDEX-Africa domain is placed at 44 ∘S, which limits considerably the window over which upper-level westerlies can be analyzed. Such an effort has been made in Karypidou, (2022) (Figures 5.4.6 – 5.4.7), however, the "landscape" over which upper-level westerlies were analyzed was extremely limited. Moreover, the CORDEX-Africa ensemble simulations do not use an ocean model coupled to the atmospheric component of the RCM; prescribed SST's only are used by RCMs. Considering that, it would be problematic to analyze westerly winds mainly blowing over the southern hemisphere oceans and sporadically crossing over land (such as southern Africa or South America, and Australia). Further explanations are provided in [3].

Karypidou, M. C.: Estimating errors and uncertainties of precipitation in regional climate simulations over southern Africa: investigation of physical processes, 2022.

3. We understand this point and find it extremely interesting concerning the impact of the Tropical-Extratropical (TE) cloud bands on precipitation over southern Africa, as analyzed in Hart et al., (2018) (with background fundamental work being described in D'Abreton and Lindesay, (1993)). As stated in Hart et al. (2018) "*The seasonality of TE cloud band likelihood emerges from the finely tuned interaction between the asynchronous seasonal cycles in subtropical upper-level westerlies and lower-tropospheric instability.*" Also, as it is stated in Howard and Washington (2018), the Angola Low pressure system can be considered as a precursor to the Tropical Temperate Troughs (TTTs), as "*The Angola low enables southward transport of atmospheric water vapor from the tropics, crucial to the development of TTTs*". We consider that the Angola low pressure system provides the necessary lower-tropospheric instability required for the formation of the TE cloud bands analyzed in Hart et al. (2018). However, since the upper-level westerly flow is available almost throughout the whole year (with variations in intensity and latitude of occurrence – Figure 5.4.5 in Karypidou, (2022)), the key agent for rainfall during the early rainy season is the occurrence of low-tropospheric instability (i.e. Angola low). Therefore, although the interplay between upper-level westerly flow and low-level atmospheric instability is crucial for the development of cloud bands, rainfall during the early rainy season can originate from alternative agents. By "alternative agents" we mean again the Angola low pressure system (at its heat low phase) and moisture supply being provided to the Angola region from low-level westerly winds. Considering the work by Howard and Washington (2019), the Congo Air Boundary is crucial in constraining moisture to the northwest part of southern Africa, necessary to fuel early season rainfall over the greater Angola region. The location of CAB significantly affects the amount of rainfall that the northwestern part of southern Africa will experience ("*the CAB plays a primary control on the spring and early summer rainfall in southern Africa. Early in the season, more rainfall may occur in this region when the CAB is farther south*"). In addition, considering all the technical constraints concerning the geographical extent of the CORDEX-Africa domain, the methodology employed in Hart et al. (2018) could not have been employed here. More specifically, Hart et al. (2018) analyze streamfunctions at 200 hPa and distinguish between eddy-driven jet axes and distinguishable jet axes over latitudes ranging up 70 ∘S. Such analysis could not have been performed in CORDEX-Africa, considering that the southernmost latitude of the CORDEX-Africa domain is 44 ∘S.

Howard, E. and Washington, R.: Drylines in Southern Africa: Rediscovering the Congo Air Boundary, J. Climate, 32, 8223–8242, https://doi.org/10.1175/JCLI-D-19-0437.1, 2019a.

Karypidou, M. C.: Estimating errors and uncertainties of precipitation in regional climate simulations over southern Africa: investigation of physical processes, 2022.

4. Concerning the land-atmosphere coupling over southern Africa, we make reference to the work by Careto et al., (2018), who investigated land-atmosphere coupling metrics within the CORDEX-Africa ensemble (hindcast simulations: ERA-Interim driven). More specifically, the Pearson correlation between the surface upward latent heat fluxes (hfls) and the sensible heat fluxes (hfss) were found to be strongly anticorrelated over the Sahel, southern Africa, and eastern Africa regions, especially during DJF, but also during SON. In Figure 6 of Careto et al., (2018), strong negative correlation values are indicative of strong land-atmosphere coupling. In addition, Careto et al., (2018) introduce the Latent Heat Flux-Temperature Coupling Magnitude (LETCM) metric, which for SON displays its highest values over southern Africa (Figure 7c) and co-occurs with the region of strong negative correlation between hflss ~ hfss (strong coupling situations). Since coupling as quantified using both metrics (correlation hflss ~ hfss and LETCM) is strongest over the Angola region during the early rainy season (SON), and since coupling in climate models is highly dependent on parameterization schemes and coupled model components simulating land processes (Wilhelm et al., 2014), we hypothesize that during the early rainy season RCMs will dominate precipitation signal over southern Africa.

Careto, J. a. M., Cardoso, R. M., Soares, P. M. M., and Trigo, R. M.: Land-Atmosphere Coupling in CORDEX-Africa: Hindcast Regional Climate Simulations, Journal of Geophysical Research: Atmospheres, 123, 11,048-11,067, https://doi.org/10.1029/2018JD028378, 2018.

Wilhelm, C., Rechid, D., and Jacob, D.: Interactive coupling of regional atmosphere with biosphere in the new generation regional climate system model REMO-iMOVE, Geoscientific Model Development, 7, 1093–1114, https://doi.org/10.5194/gmd-7-1093-2014, 2014.

**Minor Comments**

**3rd Comment:**

Line 197-198, (line 300 too). Excess surface heating is surely even greater in peak summer months? Furthermore, heating is insufficient for convection when surface environments are moisture limited as they are during October.

RESPONSE: October is characterized by a transitional weather regime namely between the dry and wet season. The sensible heat flux is surely consistently larger during summer peak months but here we point to the following:

(i) What happens in the upper levels of the atmosphere: October being a transitional month, we can expect some of the first intrusion of "colder" air in the upper pressure levels. Higher vertical gradient increases parcels buoyancy, i.e., larger CAPE and the general instability of the atmosphere which allows convective phenomena to occur also (especially) in moisture-limited contexts.

(ii) Soil moisture precipitation feedback: Several works stress how this feed back can be both positive (more precipitation initiation where is wet) or negative (more precipitation initiation where is dry). Moreover, another crucial aspect is the spatial distribution of the soil moisture in the study area but also in the surrounding areas, its "patchness", which is able to alter mesoscale circulation and precipitation pattern distribution (Seneviratne et al. 2010; Taylor et al. 2012; Graf et al. 2021). These works clearly show that moisture-limited environments not necessarily trigger negative soil moisture-precipitation feedback.

In synthesis, the upper pressure level circulation and the moisture spatial heterogeneity can trigger convective phenomena also in moisture-limited context.

Graf M, Arnault J, Fersch B, Kunstmann H (2021) Is the soil moisture precipitation feedback enhanced by heterogeneity and dry soils? A comparative study. Hydrol Process 35:. https://doi.org/10.1002/hyp.14332
Seneviratne SI, Corti T, Davin EL, et al (2010) Investigating soil moisture-climate interactions in a changing climate: A review. Earth-Science Rev 99:125–161. https://doi.org/10.1016/j.earscirev.2010.02.004
Taylor CM, De Jeu RAM, Guichard F, et al (2012) Afternoon rain more likely over drier soils. Nature 489:423–426. https://doi.org/10.1038/nature11377

**4ᵗʰ Comment:**

Line 208-209: I am not quite sure how to understand this statement about smooth topography. I read this to imply that the topography should be smooth, but is not the point of RCMs to include more detailed "jaggedness" that the real-world topography contains?

RESPONSE: Thank you for this comment! Indeed, the statement as it was initially framed is not accurate and may lead to specific misreading that is not valid for RCMs. RCMs, because of their higher horizontal resolution do represent surface characteristics such as elevation in a more accurate manner. In fact, the improvement of precipitation over the southern Africa region in the CORDEX-Africa ensemble relative to the CMIP5 GCMs was attributed to the fact that orography over the greater Tanzania region was more accurately represented in the CORDEX-Africa ensemble, blocking excess low-level moisture transport from the tropical Indian Ocean from entering mainland southern Africa (Karypidou et al., 2022), as it was the case for the CMIP5 GCMs (Munday and Washington, 2018). The comment in lines 208-209 was referring to RCA4.v1 and not all CORDEX-Africa RCMs used in the current analysis.

The following sentence has now been deleted: "*This may be attributed to the fact that the topography is not smooth enough and leads to high precipitation values over grid boxes with high elevation (Van Vooren et al., 2019).*"

Instead, the following sentence has now been placed in the text: "*This attribute is indicative of specific structural model biases related to how high-resolution elevation affects precipitation in RCA4.v1 (Van Vooren et al., 2019).*"

**5th Comment:**

Line 358, 359 linked to line 388. The only truly unambiguous signal, already well-established in literature, is this early season drying for southern Africa. So, these statements about models struggling with transition (October) seem paradoxical with the clear drying signal. In the revision of the manuscript, hopefully this can be rethought and rewritten.

RESPONSE: Thank you for this comment! Indeed, that was a point that was not stated properly and is indeed paradoxical! In the revised manuscript it has now changed to the following (the line references have now changed; however, we refer to what was prior to revision lines 358, 359, and 388.

In lines prior numbered as 358, 359, the following sentence has been omitted: *"November is the month during which there is a transition of the AL from a heat low phase to a tropical low system, and March indicates the end of the rainy season. Hence, precipitation during the transition months is challenging for both RCMs and GCMs."*

This part has now been added instead:
*"The Angola region, which encompasses the activity of the Angola Low pressure system, displays the highest wet biases with regards to mean monthly precipitation, among all subregions examined. The months with the largest wet biases (for the Angola region) is found to be November, while the month with the largest precipitation bias spread is found to be March. In all months except of October, the CMIP5 GCMs display biases that are approximately 1-1.5 mm/d wetter than the wettest CORDEX-Africa RCM ensemble members."*

Statement in line 388 has been left unchanged since it is not in contradiction to what has been stated in lines above.

The following figure has now been added to the Supplementary material as **Figure S10**. It displays monthly precipitation biases averaged over the whole southern Africa region (SAF-All) and the three subregions examined, namely the Angola region, East Coast regions, and the SAfr region.

The text in which reference is made to **Figure S10** is the following:
*"Monthly precipitation biases averaged over southern Africa (SAF-All) and the three subregions examined are displayed in **Fig. S10**."*

[Figure]

**Fig. S10** Spatial average of precipitation bias (mm/d) from RCMs and their driving GCMs over southern Africa and the three sub-regions examined.

**6th Comment:**

Line 383 talks about observation products being kept in sight. I suspect you mean, in mind. Taking this quite literally, Kendon et al 2019 made this observation uncertainty visible by display TRMM-CMORPH bias alongside other bias or change plots (panel d in Figures 2-6). Including such a figure in your manuscript would help stay true to your line 383 statement and give the reader and indicate of magnitude of changes relative to obs. uncertainty.

RESPONSE: Thank you for this comment! Yes, we mean "observation products being kept in mind". We have now added an additional figure, in which precipitation climatology for the rainy season months (Oct-Mar) is shown for: ERA5, CHIRPS, CRU, and MSWEP. In addition, we make reference to Karypidou et al., (2022), in which precipitation uncertainty is investigated in detail among five gauge based products (datasets that are derived by spatial interpolation of rain gauges and station data: CRU.v4.01, UDEL.v7, PREC/L.v0.5, GPCC.v7, CPC-Global.v1), six satellite products and ERA5 (Figures 1, 2, 3 and Figures S10, S11).

The following has now been added in the *Data* section:

*"A fact that is commonly obscured is that observational datasets are often considered as "ground truth" however, they also are subject to multiple sources of uncertainty, caused by the underlying station datasets used, the statistical algorithms employed in spatially interpolated methods or the algorithms employed in satellite rainfall products (Le Coz and van de Giesen, 2020). More specifically, over southern Africa, it was found that gauge-based products employing spatial interpolation methods displayed high uncertainty over regions where the underlying station network was scarce, mainly over the Angola region and the northern parts of SAF (Karypidou et al., 2022). In addition, it was found that this attribute was inherited by all rainfall satellite products that were using direct merging techniques with gauge-based datasets. Here, we display monthly precipitation during the historical period (1985-2005) across four observational datasets, given in* **Table 1***. More specifically, we use the CRUv4.06 dataset (Harris et al., 2020), which is a purely gauge-based product (employing station data and a spatial interpolation algorithm to provide a spatially continuous gridded product), ERA5 (Hersbach et al., 2020), which is a reanalysis product, CHRIPS (Funk et al., 2015), which is a satellite rainfall product, and finally, MSWEP (Beck et al., 2017) which is a product merging station data, satellite data and dynamic model outputs. All datasets have been analyzed using monthly mean values. The results are displayed in* **Fig. 1***. As shown, there is a substantial agreement among them both with regards to the spatial and temporal pattern of monthly precipitation over southern Africa."*

Table 1 Gauge-based, satellite, reanalysis and merged precipitation products analyzed over the study region using monthly mean precipitation for the period 1985-2005.

| Dataset | Resolution | Frequency | Type | Period |
|---------|-----------|-----------|------|--------|
| CRU TS4.06 | 0.5° | Monthly total | Gauge-Based | 1901-2021 |
| MSWEP | 0.1° | 3-hourly | Merged product | 1979-present |
| CHIRPS.v2 | 0.05° | Daily totals | Satellite | 1981-present |
| ERA5 | ~0.25° | Hourly | Reanalysis | 1979-present |

[Figure]

Figure 1. Monthly mean precipitation climatology for the period 1985-2005.

**7th Comment:**

For clarity remove region letters in text and figures. Just go with SAF-All, Angola, East Coast, SAfr, or similar throughout to make for easier reading.

RESPONSE: Thank you! We have done so in all text and figures.

**8th Comment:**

Bias plots will be much clearer to interpret if expressed in % bias from climatology (with mask for negligible rainfall areas, e.g. <1mm/month) and if colours were white for low/no bias increasing to dark red (dry) or blue (wet).

RESPONSE: Thank you for this suggestion. Indeed % bias helps to clearer image of bias, however, percent bias is tricky with very small values. Reviewer suggests we mask areas with negligible rainfall to bypass this issue. However, very large areas during October experience rainfall <1 mm/d. During November and March there are also large areas with values <1 mm/d or marginally larger than this threshold, so eventually large regions are masked from the panel maps – and the masks are variable among the months of the rainy season. For this reason, we chose to display biases as the difference model-obs.

**9th Comment:**

A paper you may have missed by which is relevant to some of your analysis here is Munday, C., & Washington, R. (2019). Controls on the diversity in climate model projections of early summer drying over southern Africa. Journal of Climate, 32(12), 3707-3725.

RESPONSE: Thank you very much for mentioning this paper! We have now included it in our analysis. More specifically, the following portion has now been added in the 5th paragraph of the *Discussion and conclusions* section (new sentences are indicated in bold):

"Concerning the climate change signal, there is a strong agreement among all GCMs and RCMs that precipitation during October will decrease by (-0.1) – (-1) mm/d, a fact associated with a projected later onset of the rainy season, which is further **linked with** a northward shift of the tropical rain belt **(Dunning et al., 2018; Lazenby et al., 2018). The topic of reduced early rainfall over southern Africa for the end of the 21st century under all emission scenarios/pathways has been examined extensively for the CMIP3 and CMIP5 GCM ensembles (Seth et al., 2011; Cook and Vizy, 2021; Lazenby et al., 2018; Howard and Washington, 2019b). A common observation in all CMIP5 GCMs for the early rainy season by the end of the 21st century is that instability over southern Africa reduces, surface temperature increases, and the heat low phase of the Angola Low pressure system is strengthened (Howard and Washington, 2019). However, rainfall decline in the CMIP5 ensemble over southern Africa should be additionally considered in the context of the systematic precipitation biases already diagnosed in the historical simulations (Munday and Washington, 2018; Howard and Washington, 2019). Considering that the systematic wet precipitation bias is significantly reduced in the CORDEX-Africa ensemble relative to their driving CMIP5 GCMs (Karypidou et al., 2022), we gain confidence that future precipitation projections according to the CORDEX-Africa ensemble provide a more plausible future scenario.** For the rest of the months, the results are variable, indicating the need for a multi-model approach, when climate change impacts are assessed. A feature that is identified in some GCMs and is transferred to the downscaling RCMs, is a precipitation increase that extends from the central SAF region towards the southeast. This result is consistent with previous work that shows an increase in frequency of landfalling cyclones along the eastern seaboard of SAF (Muthige et al., 2018). Since tropical cyclones are a particular cause of severe flooding events over the region of Mozambique, there is an urgent need for planning and mitigation strategies over the region."

**References:**

D'Abreton, P., and J. Lindesay, 1993: Water vapour transport over southern Africa during wet and dry early and late summer months. Int. J. Climatol., **13**, 151–170, https://doi.org/10.1002/joc.3370130203.

Hart, N. C. G., Washington, R., & Reason, C. J. C. (2018). On the Likelihood of Tropical–Extratropical Cloud Bands in the South Indian Convergence Zone during ENSO Events, Journal of Climate, 31(7), 2797-2817. Retrieved Mar 19, 2022, from https://journals.ametsoc.org/view/journals/clim/31/7/jcli-d-17-0221.1.xml

Kendon, E.J., Stratton, R.A., Tucker, S. *et al.* Enhanced future changes in wet and dry extremes over Africa at convection-permitting scale. *Nat Commun* **10,** 1794 (2019). https://doi.org/10.1038/s41467-019-09776-9